# Single particle cryo-EM reconstruction of 52 kDa streptavidin at 3.2 Angstrom resolution

Xiao Fan [1,2,4], Jia Wang[1,4], Xing Zhang[1], Zi Yang[1,2], Jin-Can Zhang[3], Lingyun Zhao[1], Hai-Lin Peng [3], Jianlin Lei [1,2] & Hong-Wei Wang [1,2]

The fast development of single-particle cryogenic electron microscopy (cryo-EM) has made it more feasible to obtain the 3D structure of well-behaved macromolecules with a molecular weight higher than 300 kDa at ~3 Å resolution. However, it remains a challenge to obtain the high-resolution structures of molecules smaller than 200 kDa using single-particle cryo-EM. In this work, we apply the Cs-corrector-VPP-coupled cryo-EM to study the 52 kDa streptavidin (SA) protein supported on a thin layer of graphene and embedded in vitreous ice. We are able to solve both the apo-SA and biotin-bound SA structures at near-atomic resolution using single-particle cryo-EM. We demonstrate that the method has the potential to determine the structures of molecules as small as 39 kDa.

[1] Ministry of Education Key Laboratory of Protein Sciences, Beijing Advanced Innovation Center for Structural Biology, Beijing Frontier Research Center of Biological Structures, School of Life Sciences, Tsinghua University, Beijing 100084, China. [2] Tsinghua-Peking Joint Center for Life Sciences, Tsinghua University, Beijing 100084, China. [3] Academy for Advanced Interdisciplinary Studies, Peking University, Beijing 100871, China. [4]These authors contributed equally: Xiao Fan, Jia Wang Correspondence and requests for materials should be addressed to J.L. (email: jllei@tsinghua.edu.cn) or to H.-W.W. (email: hongweiwang@tsinghua.edu.cn)

With recent technical breakthroughs, cryogenic electron microscopy (cryo-EM) has rapidly become one of the most powerful and efficient technologies to investigate the structures of macromolecules at near-atomic resolution. Of the various cryo-EM structural determination methods, single-particle analysis (SPA) has drawn the most attention from structural biologists because of its relatively well-established methods for specimen preparation, data collection, image processing, and structural determination[1–3]. Thanks to the significant improvements in the recording speed and detective quantum efficiency of the direct electron detection cameras, more information at both low and high resolutions can be recovered from raw movie stacks, thus improving the reconstruction accuracy[4]. New algorithms based on Bayesian statistics have also greatly improved the efficiency of extracting signal from noisy micrographs and heterogeneous datasets[5–9]. Currently, it has become increasingly routine to reconstruct a well-behaved macromolecule with good homogeneity, rigidity, and random orientations in ice and a molecular weight larger than 300 kDa at ~3 Å resolution. In contrast, it remains a challenge to solve high-resolution structure of proteins with a smaller molecular weight, especially those below 100 kDa, using SPA Cryo-EM. The major hurdle lies in the weak contrast of the small-sized molecules embedded in vitreous ice using conventional transmission electron microscopy (CTEM). Another major obstacle remaining in SPA cryo-EM is the adsorption of proteins to the air–water interface (AWI) of the thin layer of solution during the cryo-specimen preparation[3,10,11]. Until now, the smallest protein resolved by CTEM using SPA at near-atomic resolution is the 2.9 Å resolution structure of the 64 kDa methemoglobin[12,13].

Recent hardware developments have introduced to cryo-EM new electron optical apparatuses, including energy filter, Cs-corrector, and Volta phase plate (VPP), to further improve the imaging quality. The VPP can introduce an extra phase shift to the contrast transfer function (CTF) of the objective lens, thus increasing the low-frequency signal of weak-phase objects such as frozen-hydrated biological molecules[14–16]. With new algorithms supporting the CTF determination and correction of micrographs taken with VPP[6,17,18], it was shown that VPP can be used to study various structures at near-atomic resolution, including the 64 kDa hemoglobin at 3.2 Å resolution[19–22]. Using a combination of VPP and the Cs-corrector, we demonstrated that the structure of apo-ferritin can be solved at near-atomic resolution in both under- and over-focus modes of the objective lens[23].

In this work, we use SPA cryo-EM with VPP and Cs-corrector to determine the structure of SA with a molecular weight of ~52 kDa. Different from hemoglobin that consists mostly of α-helices, SA is constituted by mainly β-strands. Our work demonstrates that the VPP can be used in SPA to resolve SA in both the apo-state and biotin-bound state at near-atomic resolution. We also find that graphene films can serve as good supporting materials to keep the SA in multiple orientations for the high-resolution structural determination. Our results prove in principle the capability of SPA cryo-EM to solve the atomic models of small-sized proteins and their ligand-bound complexes. This development would be of potential application in structure-based drug discovery.

## Results

**Preparation of frozen-hydrated SA specimen on graphene film.** In this study, we used a single-crystalline monolayer graphene over a Quantifoil R0.6/1 gold grid as the supporting film to facilitate the cryogenic SA specimen preparation (see Methods for more details). Using a modified version of our previous imaging strategy to combine the Cs-corrector and VPP for cryo-EM, we were able to collect high-resolution datasets of vitrified SA specimens with high efficiency (Methods). When examined under the VPP-Cs-corrector-coupled Titan Krios at 300 kV with phase shift ranging from 30° to 120°, the SA specimens demonstrated monodisperse particles with a high contrast that could be easily identified and picked using automatic algorithms (Fig. 1a). We found that the single-crystalline graphene with a monolayer of carbon atoms introduced very low background noise to the specimen and could also serve as a good reference for the assessment of the cryo-EM image quality and motion correction with its hexagonal lattice signal[24–26]. After the motion correction of the raw movie stacks of the specimen, we calculated the Fourier transform of the motion-corrected micrographs. In micrographs with good quality, we observed clear reflection spots at 2.13 Å resolution in a hexagonal pattern corresponding to the graphene lattice at its first order (Supplementary Fig. 1), indicating a successful motion correction with high-resolution information recovered to at least 2.13 Å. It is worth noting that these reflection spots were not clear or sharp enough without the proper motion correction (Supplementary Fig. 1). Therefore, the sharpness of the reflection spots of single-crystalline graphene in the Fourier transform can serve as a good indicator to judge the quality of the micrographs and the motion-correction efficiency. We also examined the Fourier transforms of various areas on the same specimen grid and found that most of them demonstrated a consistently hexagonal lattice diffraction pattern in the same orientation, indicating the presence of a single-crystalline graphene film over the grid (Supplementary Fig. 1D).

**Single-particle reconstruction of SA by VPP-cryo-EM.** Using the automatic particle picking algorithm Gautomatch, we extracted ~710,000 and 1,350,000 particle images from the good motion-corrected micrographs of SA in the absence and presence of biotin, respectively, and applied a 120 Å Fourier high-pass filter to the particles prior to further processing (Supplementary Fig. 2). The high-pass filter turned out to be necessary for the correct alignment of the particle images (Supplementary Fig. 2), probably reducing the low-frequency background bias, in agreement with our previous results[27]. Reference-free two-dimensional (2D) alignment and classification from such datasets yielded 2D class averages with clear secondary structural features that matched the atomic model of the SA protein (Supplementary Fig. 2B and 2C). Using an initial model generated de novo by the Stochastic Gradient Descent (SGD) method in Relion[6], we performed multiple rounds of three-dimensional (3D) classifications to screen the best particles for the final 3D refinement and reconstruction (Supplementary Figs. 3 and 4). In the end, we obtained a reconstruction of apo-SA at 3.3 Å resolution (with D2 symmetry applied during the refinement, Fig. 1c) from a final dataset composed of ~24,000 particles (Supplementary Table 1) and a reconstruction of the SA–biotin complex at 3.2 Å resolution (with D2 symmetry applied during the refinement, Fig. 1d) from a final dataset comprising ~45,000 particles (Fig. 1e, Supplementary Table 1). We also performed reconstructions of the two different states without imposing any symmetry (Supplementary Fig. 5A). These reconstructions have a very similar map quality to those calculated with the D2 symmetry, albeit with slightly lower resolutions (Supplementary Fig. 5C).

The 3D reconstructions of SA in its apo- and biotin-bound states were both clear enough to depict all the secondary structural elements and most of the side chains (Figs. 2 and 3, Supplementary Movie 1). The atomic model of SA solved previously by X-ray crystallography (PDB 1MEP[28]) can fit into the EM densities with a correlation coefficient ~0.74, indicating the structural fidelity of SA in its crystallographic and soluble

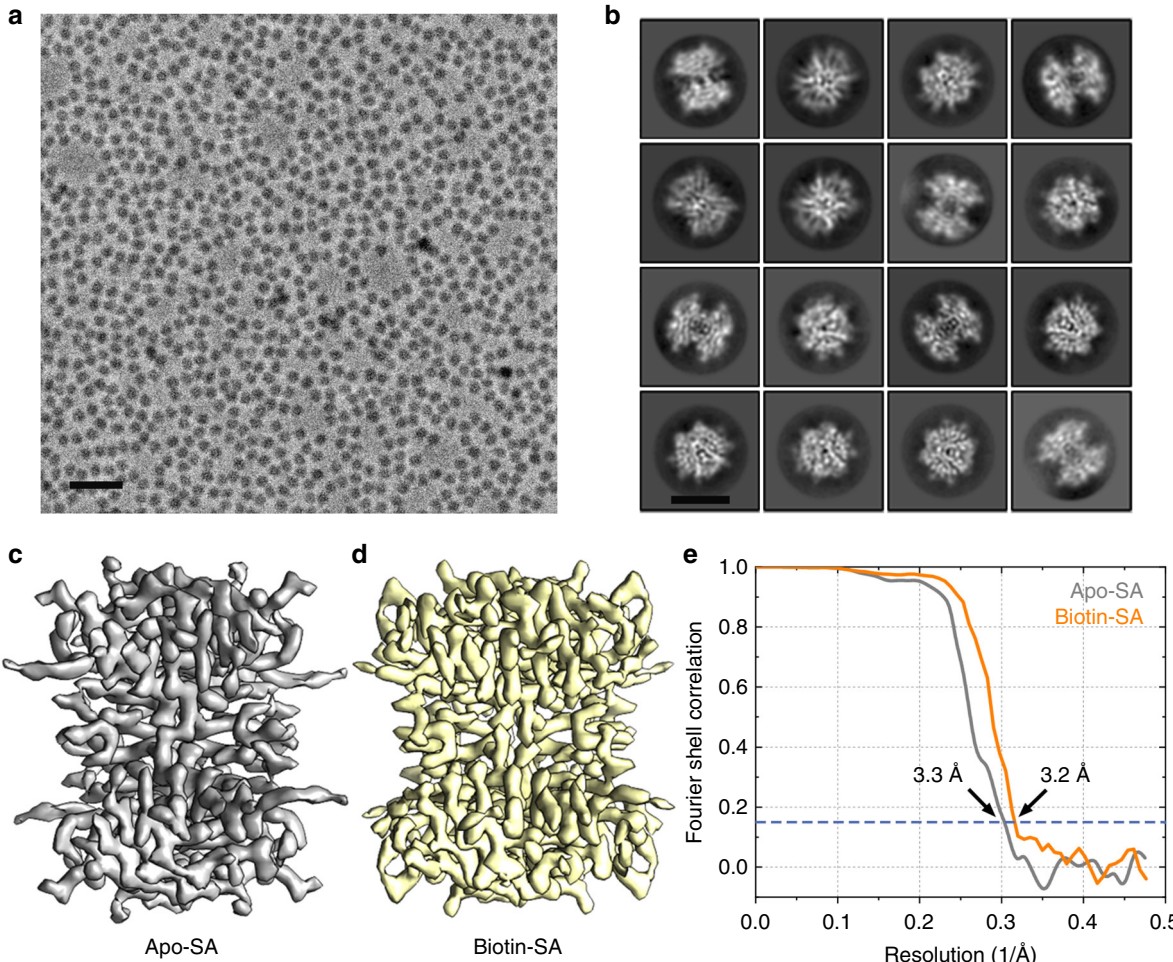

**Fig. 1** The SPA cryo-EM of SA. **a** A representative micrograph of the SA specimen by the VPP-Cs-corrector-coupled cryo-EM. The scale bar represents 20 nm. **b** Representative 2D class averages of SA particle images. The scale bar represents 5 nm. **c** The 3D reconstruction of apo-SA at 3.3 Å resolution from 23,991 particles and **d** the 3D reconstruction of the biotin-bound SA at 3.2 Å resolution from 45,686 particles. **e** The fourier shell correlation (FSC) curves of the two reconstructions using the gold-standard criteria

forms. The density of biotin in the SA–biotin reconstruction can be precisely identified with the unambiguous docking of biotin's atomic model (Fig. 2). Compared with the biotin-bound SA, the density corresponding to loop 46–51 in the EM map of apo-SA was missing (Fig. 2), indicating that this lid-like loop is flexible without ligand binding. In contrast, this loop can be clearly defined in the EM map of biotin-bound SA, in which the major side chains (ASN23, SER27, TYR43, ASN49, and SER88) forming a stable hydrogen bond network around the biotin ligand are well resolved (Fig. 2).

**Focused classification analysis of the biotin-binding pocket**. A critical problem in drug discovery is to identify the ligand-binding site of target proteins. We wondered whether the ligand-binding site could be determined via image processing in small proteins such as SA without prior knowledge[29–33]. As SA is a tetramer and has four biotin-binding sites in each protein, we treated each SA monomer (with one binding pocket) as an asymmetric unit and used the angular information from the reconstruction with D2 symmetry to align the four asymmetric units from the same particle to a given orientation. This step generated a dataset four times larger and comprising roughly aligned asymmetric particles, thus called the asymmetric particle dataset. After a local search refinement with C1 symmetry, the

asymmetric particle dataset was subjected to 35 iterations of 3D classification into 4 classes in a skip-alignment mode in Relion. Without specifically focusing on the binding pocket, a soft mask slightly larger than the SA monomer was applied in either the refinement or classification. We performed this 3D skip-alignment classification analysis of the apo-SA and biotin-SA datasets separately, and found a rather small occupancy variance around the biotin-binding pocket among the different classes in each dataset (Supplementary Fig. 6A and 6B), demonstrating unambiguously the lack of biotin in all the monomers of apo-SA and the full occupancy of biotin in all the monomers of biotin-SA. This result occurs because SA has a very strong binding affinity to biotin and the condition of the biotin-SA specimen allowed the full occupancy of the protein's ligand-binding sites. The ligand occupancy, however, may not be full for other proteins and other conditions. Thus, we tried to test whether we could extract the ligand-binding information by image processing from particles with a partial ligand occupancy. We mixed the apo-SA and biotin-bound SA datasets, and analysed them as one dataset for the 3D refinement. The reconstruction of the mixed dataset demonstrated a structure showing a biotin-like density in the binding pocket at 3.1 Å resolution (Fig. 4a, b, Supplementary Fig. 5B). From this mixed dataset of apo-SA and biotin-SA, the 3D skip-alignment classification of the asymmetric unit into four

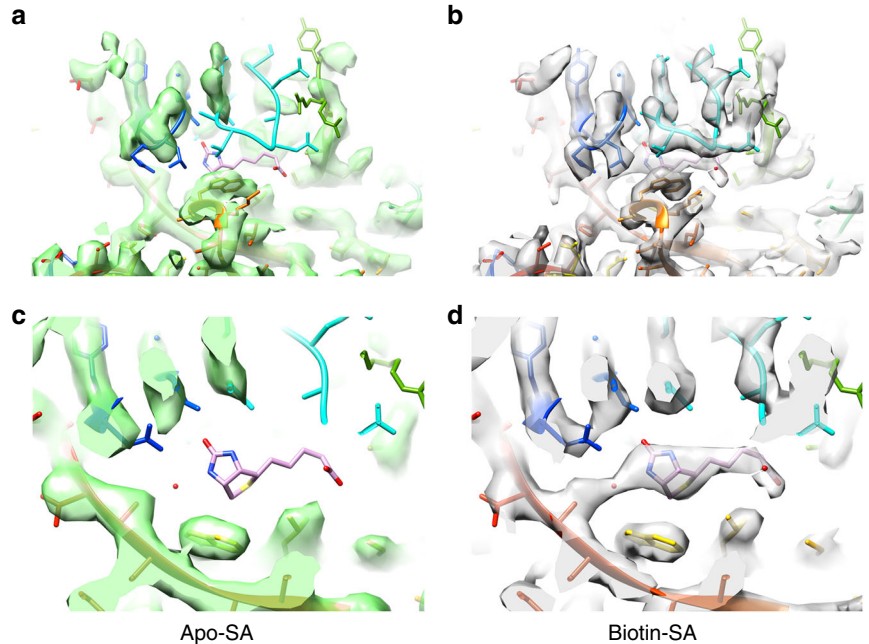

Apo-SA                    Biotin-SA

**Fig. 2** Comparison between the reconstructions of the two SA states. **a**, **c** The region around the biotin-binding pocket of the apo-SA EM map has an empty density of the pocket and a missing loop 46–51 density, whereas in **b**, **d** the biotin-bound SA EM map, these two densities are well resolved with the atomic model of the biotin ligand and the loop 46–51

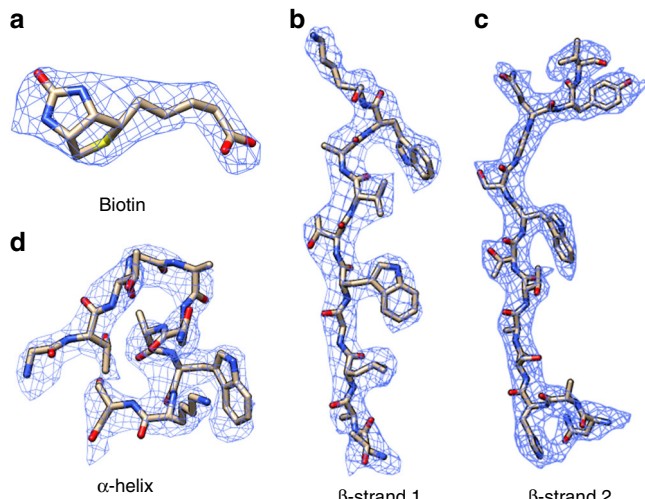

Biotin

α-helix              β-strand 1          β-strand 2

**Fig. 3** Biotin-SA local maps with their corresponding atomic models. **a** Biotin density in the binding pocket. **b** Representative densities of secondary structures: β-sheet (**b**, **c**) and α-helix (**d**)

classes illustrated distinct differences in the biotin-binding pocket (Fig. 4c). Although Class II was vacant of biotin, the other three classes all had partial biotin occupancy in the binding pocket. We further refined the 3D reconstructions using particles in Class II (Fig. 4d) or the merged Class I–III–IV (Fig. 4e) individually. The refined 3D maps showed more clearly that the Class II reconstruction lacked a density corresponding to loop 46–51 and the biotin ligand (Fig. 4d, red circle), whereas the Class I–III–IV reconstruction maintained both clearly (Fig. 4e, blue circle).

We further randomly split the biotin-SA dataset into 20 subsets (9140 monomers in each subset) and mixed different numbers of them with the apo-SA dataset to generate 20 mixed datasets with different ratios of biotin-SA/apo-SA. We then performed the 3D refinement of these mixed datasets. As the biotin-SA/apo-SA ratio

increased, the density of loop 46–51 and biotin molecules in the reconstructions became clearer and was recognizable when the ratio was higher than 0.5 (Supplementary Fig. 6C). From the mixed dataset of M5 with a biotin-SA/apo-SA ratio of 0.5, we could further classify it to separate the biotin-bound SA structural features (Supplementary Fig. 6D). The results above implicated the capability of the heterogeneity analysis for the ligand-binding detection of proteins as small as SA by single-particle cryo-EM.

**Reconstruction of sub-tetrameric SA from subtracted dataset.** Although the 52 kDa SA is the smallest protein that has been resolved at near-atomic resolution using SPA cryo-EM until now, we were wondering whether SPA cryo-EM is capable of reconstructing even smaller proteins. We used the particle segmentation and subtraction algorithms[34,35] that are currently available in Relion to generate monomeric (13 kDa), dimeric (26 kDa), and trimeric (39 kDa) SA datasets from raw biotin-SA datasets in silica (Fig. 5a). The subtracted SA datasets had smaller molecular weights and broke the intrinsic D2 symmetry of SA and therefore the signal for the proper alignment was even weaker.

We first tested whether there was enough signal in the subtracted dataset for 2D classification with the correct angular information. The angular information of each subtracted particle was calculated in accordance with its relative orientation in the original tetrameric SA particle as well as the angular information of that tetrameric SA image in the final tetramer reconstruction. When using the correct angular information without alignment, all three subtracted datasets generated good 2D class averages with correct shapes and features (Fig. 5b, Skip Align). By removing all the angular information, we performed the reference-free 2D alignment and classification of the three datasets from scratch in Relion. In this procedure, the 13 kDa monomeric dataset generated 2D class averages with roughly correct outlines but much noisier features than the perfectly aligned controls in different views (Fig. 5b, Search Align, left panel), suggesting more alignment error in the reference-free

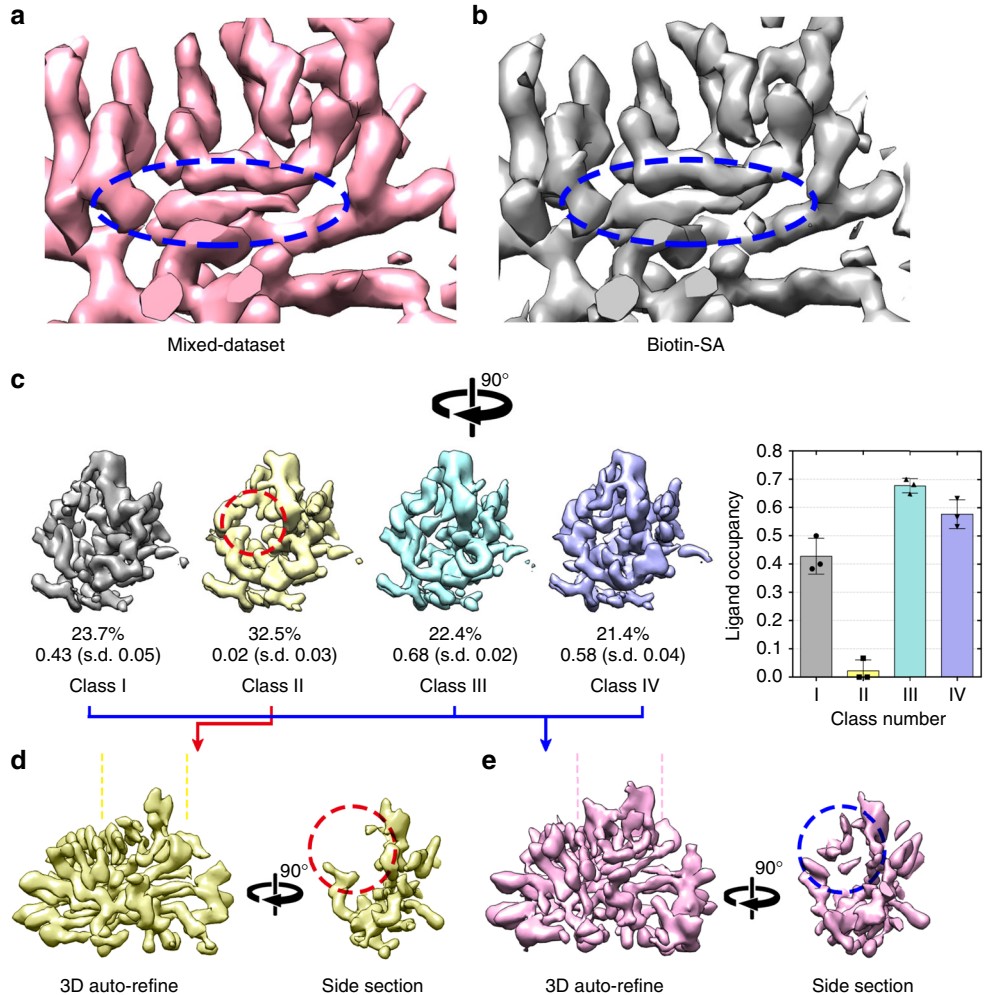

**Fig. 4** Reconstruction and classification using the mixed dataset. **a** The 3D reconstruction of the mixed dataset (apo-SA + biotin-SA) with 3.1 Å resolution demonstrates a biotin-bound-like density (circled in blue) as **b** the 3D reconstruction of biotin-SA at 3.2 Å resolution of a monomer. **c** Asymmetric 3D classification reconstructions of the mixed dataset. The empty density of the biotin-binding pocket in the monomer of the Class II reconstruction is circled in red in contrast to the ligand densities in the other classes. The percentage of particles and ligand occupancy in each class are labeled. A column graph with error bars to show the ligand occupancy of each class is shown. The 3D reconstruction of **d** class II and **e** the merged class I–III–IV indicated the apo-SA and biotin-SA individually. A side section comparison demonstrated the extra density of loop 46–51 and the biotin molecule in **e** (blue circle). Error bars (SD) were calculated from three random repeats. Source data are provided as a Source Data file

alignment. The 26 kDa dimeric dataset generated one well-aligned view (Fig. 5b, Search Align, middle panel), whereas the other views were misaligned. The 39 kDa trimeric dataset generated correct shapes and features in multiple views (Fig. 5b, right panel as representatives), indicating a successful reference-free alignment.

To verify whether the subtracted datasets can still generate valid 3D reconstructions, we used the correct angular information to perform local 3D refinement. Indeed, given the correct angular information, all the three subtracted datasets yielded correct reconstructions (Fig. 5c). We further tested whether the images from those three datasets had enough signals to search for the correct angular information without any prior knowledge. The 39 kDa trimeric dataset had enough signals to generate a correct 3D reconstruction via a global angular search from scratch (Fig. 5c). In contrast, the monomeric and dimeric datasets failed to reconstruct high-resolution structures in global refinement (Fig. 5d), probably due to the lack of sufficient signals to align.

The 3D refinement results of the three datasets were consistent with the 2D classification, indicating that: (1) all datasets contained enough signals for reconstruction at high resolution if the angular information is correct and (2) the 39 kDa trimeric SA images already contained enough signals for image processing from scratch to obtain a high-resolution structure. The results also indicated that good 2D class averages with clear features would provide a high possibility of successful reconstruction. In our results, the 26 kDa dimeric dataset could generate high-quality 2D class averages of certain orientations but not all of the views. The lack of accuracy of the alignments in the other orientations probably caused the failure of its 3D refinement. We infer that the major constituents of the β-strands in SA made the alignment difficult in some orientations. Nevertheless, the successful reconstruction of the trimer dataset indicated the capability of solving an asymmetric protein structure with a molecular weight ~39 kDa at near-atomic resolution by SPA cryo-EM.

**Distribution of SA particles in the vitrified specimen.** We noticed that even after the careful scrutiny of the SA particle images by 2D classification to remove all obvious junk or bad particles, only ~20% (79,289 vs. 378,987 for the apo-SA, Supplementary Fig. 3) of the seemingly good particles contributed to

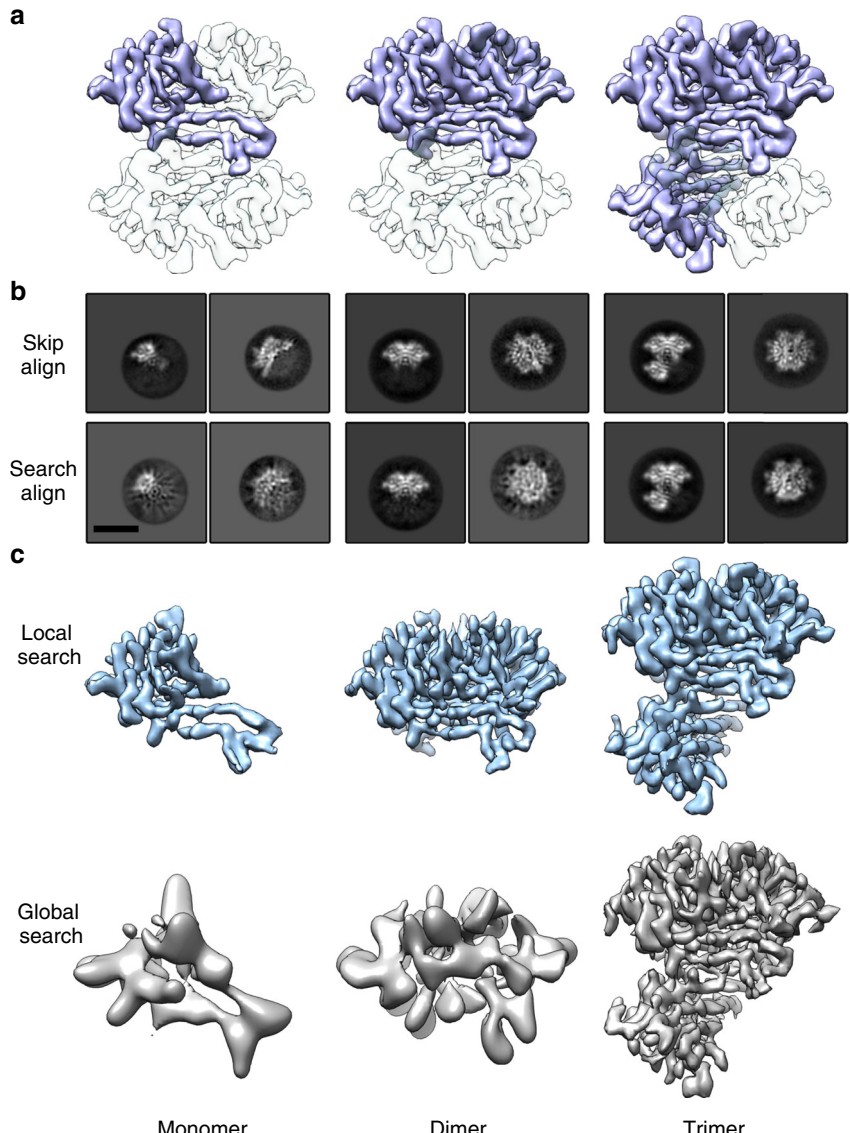

**Fig. 5** Reconstructions of subtracted SA in different oligomeric states. **a** The diagrammatic sketch of the subtraction in raw biotin-SA particles. The white parts were subtracted from the individual particle images based on related angular information, with the blue part left for image processing (monomer, dimer, and trimer from left to right). **b** 2D classification results from subtracted datasets in different oligomeric states either using the angular information generated from the 3D refinement of the SA tetramers (Skip Align) or omitting the angular information in a reference-free mode (Search Align). The scale bar represents 5 nm. **c** 3D reconstructions with local angular search of the three subtracted datasets. The initial rough angular information was generated from the 3D refinement of the SA tetramers. **d** 3D reconstructions with global angular search. The results indicate that only the trimeric dataset could yield a successful global refinement from scratch

the correct high-resolution reconstruction after the 3D classification. Indeed, despite our various efforts on image processing procedures, the other 80% of the particle images did not generate reconstructions with clear secondary structural details, even though they appeared very similar in our eyes to the good particles for the high-resolution reconstructions. We confirmed that the original micrographs containing these particles were of high quality.

We set out to investigate what made the difference for the particles to contribute to the high-resolution reconstruction. It has been hypothesized that the adsorption of protein molecules on the AWI may cause the denaturation or partial unfolding of the protein[3,10,11]. We were wondering whether the location of the particles in the thin layer of vitreous ice caused the variation of the image quality for the high-resolution reconstruction. We

therefore performed the electron tomography of the same grid for the single-particle data collection of SA on the graphene-supporting film using VPP-Cs-corrector-coupled cryo-EM. The 3D reconstructions of the tomograms were clear enough for us to depict the SA particle distribution in the specimen (Supplementary Movies 2 and 3, Fig. 6, and Supplementary Fig. 7). It is interesting to see that the SA particles distributed mainly in two different layers along the z-direction, one on the AWI and the other on the graphene–water interface (GWI) (Fig. 6a, Supplementary Fig. 7). There were very few particles between these two layers. This observation suggests that during the specimen preparation, the SA molecules either stuck to the GWI or got adsorbed onto the AWI. Surprisingly, the particles on the GWI had an uneven distribution, mostly in clustering areas (Fig. 6a, red arrow) and only a few in lacuna areas (Fig. 6a, blue arrow). In

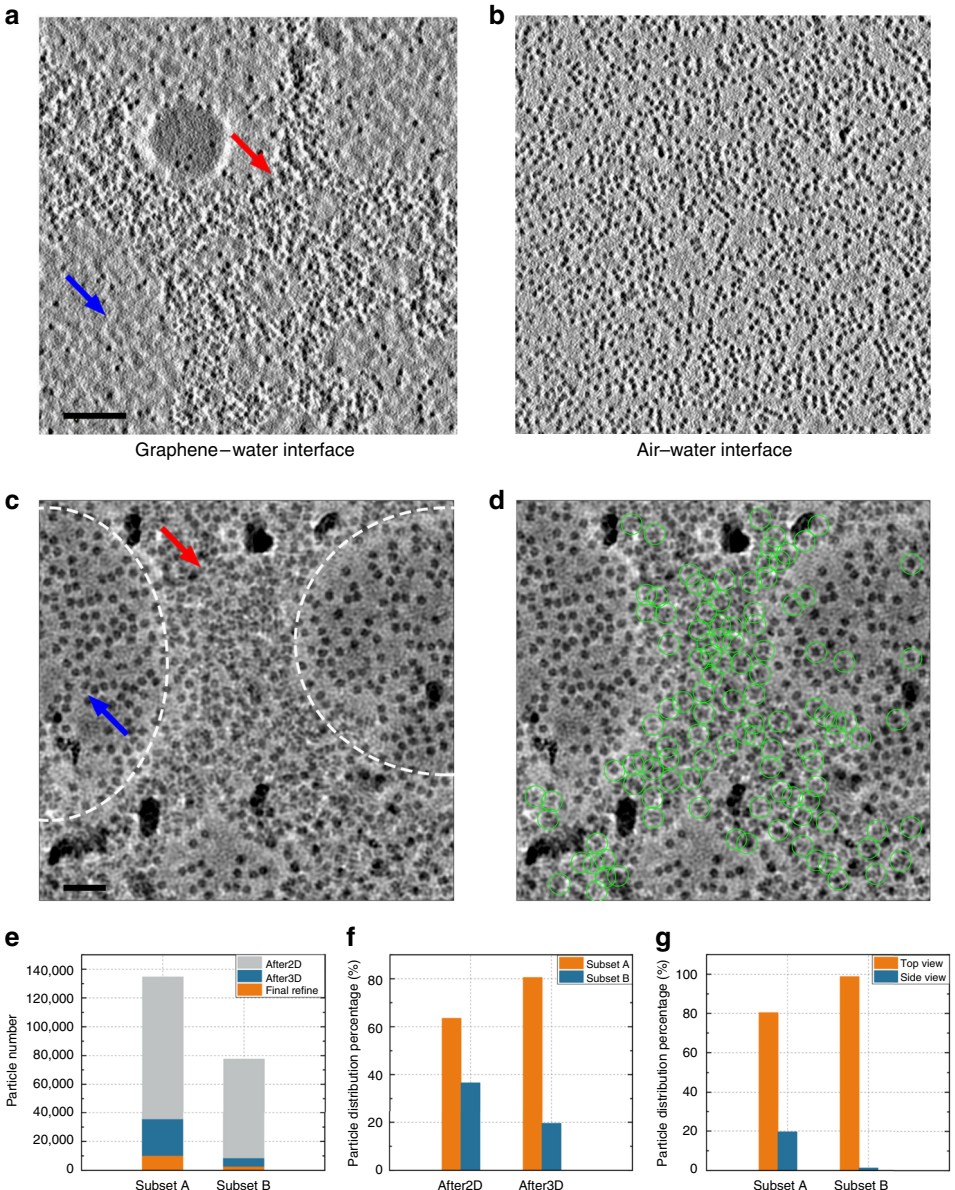

**Fig. 6** SA particles on graphene–water and air–water interfaces. **a** The *X–Y* cross-section corresponding to the graphene–water interface (GWI) from a reconstructed tomogram. The uneven distribution of particles is indicated as a clustering area (red arrow) and lacuna area (blue arrow). The scale bar represents 100 nm. **b** The *X–Y* cross-section corresponding to the air–water interface (AWI) from the same reconstructed tomogram as in **a**. **c** A micrograph containing a clustering area (red arrow) and uniform distribution area (blue arrow). The boundaries of the two areas are marked by dashed lines. The scale bar represents 20 nm. **d** The same micrograph as in **c**, with the particles that contributed to the final high-resolution reconstruction circled in green. **e** The numbers of particles in Subset A and Subset B after the 2D Classification (After2D), after the first 3D Classification (After3D), and in the final Refinement (Final Refine) were counted in 749 selected micrographs with a clear clustering feature. **f** The percentages of particles from Subset A and Subset B in the different data processing steps. **g** The distribution of the major particle orientations, top view (rounded-like) and side view (butterfly-like), in Subset A and Subset B, respectively. The particles in Subset B demonstrated a severe preferential orientation. Source data are provided as a Source Data file

contrast, the particles on AWI showed a more uniform and dispersed distribution (Fig. 6b). Such a phenomenon was observed in both relatively thick (~50 nm, Supplementary Fig. 7C, Supplementary Movie 2) and thin (~10 nm, Supplementary Fig. 7D, Supplementary Movie 3) ice. The electron tomography analysis implied that the micrographs of SA single particles collected at a zero-degree tilt actually reflected the superposition of the particles on both GWI and AWI.

We analysed all the micrographs that were used for the single-particle reconstruction by sorting them by the percentage of good particles classified in the correct high-resolution reconstruction.

We found that the best micrographs with the highest percentages of good particle images had uneven particle distributions (Fig. 6c), and that the good particles contributing to the correct high-resolution reconstruction came mostly from the clustering areas with a similar pattern to those on the GWI, as revealed by electron tomography (Fig. 6d). The particles in the more uniform and dispersed areas contributed much less to the final high-resolution reconstruction.

Intrigued by the observation of the particle distribution, we went through the entire apo-SA dataset to verify the potential correlation between the particle distribution on the grids and

their contribution to the high-resolution reconstructions. Out of the 1450 micrographs of apo-SA, we manually selected 749 micrographs with clear features of particle clustering and extracted 212,105 particles from these micrographs using the particle position information calculated from the previous reference-free 2D alignment and classification of the entire apo-SA dataset. Based on the location of each particle, the 212,105 particles were manually divided into two subsets with 134,606 particles in the clustering regions (subset A) and 77,499 particles in the uniformly dispersed regions (subset B) (Fig. 6a–d). Ideally, such a division should put most of the particles on the GWI in subset A and leave subset B with mainly particles on the AWI. We subsequently correlated the particles in the two subsets to the particles in the different steps during the 3D classification and refinement of the entire apo-SA dataset (Supplementary Fig. 3). This assigned 44,326 particles from the 749 micrographs that contributed to the 3.5 Å resolution reconstruction after the first round of 3D classification. We immediately noticed that among the 44,326 particles, 35,676, accounting for >80%, were from subset A and only 8,650, accounting for <20%, were from subset B (Fig. 6e, f, Supplementary Table 2). It is also worth noting that the percentage of particles retained in the best 3D class from subset A is 26.5% (35,676/134,606; Supplementary Table 2), higher than that from subset B, 11.1% (8,650/77,499; Supplementary Table 2). The fact that the contribution of the particles from subset A to the best 3D class increased from 63.5% (134,606/212,105) to 80.5% (35,676/44,326) after the classification indicates that a larger portion of the molecules on the GWI were well-preserved.

To further understand the difference between the two subsets of particles, we performed a reference-free 2D classification on them and found that the subset B particles exhibited more severe preferential orientations than those of subset A (Fig. 6g, Supplementary Fig. 8A, B). We also compared the 3D reconstructions of the particles in subsets A and B, either using angular information from the previous 3D refinement of the entire apo-SA dataset or by recalculating them from scratch. The maps from subset A consistently demonstrated the correct features of the SA molecule, whereas the maps from subset B were of poor quality (Supplementary Figs. 8C, D, 9).

To further investigate the effect of AWI on the structure of SA molecules, we performed cryo-EM analysis of apo-SA on regular holey carbon grids without graphene support. SPA of these SA molecules demonstrated a strong preferential orientation, which is similar to that of subset B in the above analysis (Supplementary Fig. 10).

## Discussion
As SPA cryo-EM has become a powerful method for solving structure of supramolecular complexes with a large molecular weight, the question of how small a molecule can be solved at a near-atomic resolution by this method has drawn more attention. Here we demonstrated that using VPP and Cs-corrector, SPA cryo-EM can solve SA, with a molecular weight of ~50 kDa, at ~3 Å resolution, good enough to determine the ligand-binding site. By combination with particle subtraction analysis, we could push the lower boundary of the molecular weight further to at least a 39 kDa asymmetric tetramer at ~3 Å resolution. Although a 3 Å resolution reconstruction is not sufficient to accurately assign every atom from a specific small molecule, with additional information on the possible conformations of the molecule, we could identify the binding pocket and possible interactions (such as hydrogen bonds) between the protein and ligands. More recent progress in algorithm development has enabled the solution of macromolecules with a large molecular weight and

high symmetry at better than 2 Å resolution[36–38]. Previous theoretical predictions suggested that SPA could determine the atomic structure of proteins with a molecular weight as small as ~20–40 kDa[39–41]. It is foreseeable that macromolecules as small as streptavidin or even smaller can be solved at a high-enough resolution to build atomic models of the ligand de novo. Such a scenario would make the cryo-EM structure-based drug discovery more trivial. More importantly, the unique power of single-particle cryo-EM in dealing with heterogeneous ligand occupancy and conformations in a single specimen could help accelerate the drug-screening process without the need for crystallization trials or crystal soaking, which are both time- and material-consuming. Our focused classification approach to quantify the ligand density of the dataset suggested a limited capability to obtain a full occupancy of the bound ligand. This is likely to be due to the low signal to noise ratio (SNR) of the images and the current algorithm's limitation. With the developments of hardware and software in the future, one may solve the structures of a target macromolecule co-existing with multiple ligand candidates in the same specimen and obtain multiple ligand-occupied states solved simultaneously within a few days of data collection and computation.

In this work, we used the Cs-corrector in combination with the VPP for data acquisition, which has been demonstrated by us to allow imaging at the under-focus or over-focus of the objective lens for cryo-EM[23]. However, our results in this work do not suggest a necessity of the Cs-corrector for the high-resolution structural determination of small protein complexes. As shown in the works by Khoshouei et al.[19] and Herzik et al.[13], the 64 kDa hemoglobin can be solved at ~3 Å resolution without the Cs-corrector or VPP.

This work solves a near-atomic resolution structure of a protein smaller than 100 kDa with a supporting film. Using single-crystalline graphene as the supporting film may bring us the following benefits, at least for the single-particle studies of SA: (1) reducing the ice thickness (ice noise) without introducing a strong background noise (Supplementary Fig. 7C, 7D) and (2) attracting protein particles near the GWI and reducing the adsorption onto the AWI. We also noticed that the orientation distributions of apo-SA and biotin-SA were slightly different. This difference is probably an effect on the surface property of GWI by 4 mM free biotin molecule in the biotin-SA solution. In the total two apo-SA grids and four biotin-SA grids prepared in two batches 40 days apart from each other, we found that the distribution of particles in the specimens and orientations, as well as the image quality were reproducible in our hands.

In agreement with other studies[11,42], we found that the SA particles in our frozen-hydrated specimens either stay on the GWI or become adsorbed onto the AWI, but very few stay in the bulk of the ice. This finding suggests that during the period of specimen preparation, the SA molecules move quite fast into the two interfaces and probably do not come back into the liquid bulk once hitting the interfaces[43,44]. The uneven distribution of SA particles on the GWI indicates a non-uniformed interfacial property of the graphene surface in our experiments. We do not have a good explanation for this phenomenon, whereas possible causes might include heterogeneous hydrophilicity, adsorbates contamination, or more complicated graphene–water interfacial interactions on the graphene surface[45,46]. Taking advantage of the different distribution patterns of SA particles on the GWI and AWI, we could roughly separate the dataset in two groups. Our current results suggest that the particles staying on the GWI were better preserved with their high-resolution structural features, whereas those adsorbed to the AWI were preferentially oriented. To prevent the proteins from hitting the AWI too fast, it may be helpful to reduce the Brownian motion rate. This process could

be achieved by reducing the temperature of the protein solution, increasing the viscosity of the solution, or increasing the thickness of the liquid layer over the holes on the EM grid. However, these may all unavoidably reduce the contrast of the molecules in the cryo-EM. An ultimate solution to prevent macromolecules from hitting the AWI is by blocking the AWI with either a supporting film such as the graphene or some inert surfactant that does not have any impact on the macromolecules' structures, as suggested by Glaeser and colleagues[10,47,48]. Anchoring the macromolecules with certain affinity tags to the graphene or other electron-transparent supporting materials would be an alternative solution[49]. Only when we prevent the denaturation of the macromolecules at the AWI can we take full advantage of the single-particle cryo-EM analysis in deciphering the distribution of molecular machines in their conformational landscape beyond the static atomic models.

## Methods

**Cryo-EM sample preparation**. For the biotin-free apo-SA cryo-sample, 1 mg ml$^{-1}$ commercially available streptavidin solution (New England Biolabs) was diluted to 0.2 mg ml$^{-1}$ in 25 mM Tris-HCl buffer (pH 7.5, 75 mM NaCl). After centrifugation (12,000 × $g$, 15 min), a 4 µl diluted protein sample (0.2 mg ml$^{-1}$) was added to a pre-glow-discharged 300 mesh Quantifoil Au R0.6/1 graphene-coated grid for specimen preparation in a Vitrobot Mark IV (FEI Company). The graphene-coated grids were prepared as previously described[25]. Briefly, a large-area single-crystalline graphene grown on a copper foil produced by the chemical vapor deposition method was transferred to a Quantifoil Au holey carbon grid using a polymer-free transfer method with isopropanol solution. The copper foil was etched off by the (NH$_4$)$_2$S$_2$O$_8$ aqueous solution and washed away completely to generate a highly clean single-crystalline graphene supporting film on the Quantifoil Au holey carbon grid. Immediately before making the cryo-EM specimen, the graphene-coated grid was glow-discharged for 10 s at a low level in a Harrick Plasma instrument after its chamber was evacuated for 2 min from air. In the Vitrobot Mark IV, the humidity was set at 100%, the protein solution was applied to the grid and there was a wait of 10 s before blotting. A blot force of −1 and blot time of 1 s were applied to blot the grid after waiting. After blotting, the grid was plunged into pre-cooled liquid ethane at a liquid nitrogen temperature.

For a biotin-bound SA cryo-EM specimen, the same streptavidin solution as above was supplemented with biotin (Sigma-Aldrich, St Louis, MO, USA) to a final concentration of 0.2 mg ml$^{-1}$ SA and 4 mM biotin. The biotin-SA solution was incubated on ice for 1 h and then centrifuged at 12,000 × $g$ for 20 min. When preparing the cryo-sample, the waiting time before blotting was 2 s and the blot force was −2. The rest of the steps were the same as those in the preparation of the apo-SA specimen.

**Data collection on VPP-Cs-corrector-coupled EM**. All the data were collected on the same 300 kV Cs-corrected Titan Krios microscope, which is equipped with an FEI Volta phase plate (FEI) and a K2 Summit direct electron detector with GIF Bio-Quantum Energy Filters (Gatan). After the cryo-specimens were loaded into the microscope, we first performed the basic alignment of the microscope. Then, we tuned the Cs-corrector and VPP at eucentric-focus with approximately −0.6 µm defocus from the eucentric height at ×195,000 magnification (TEM mode, micro-probe) using a previously published procedure[23]. The microscope with a well-tuned Cs-corrector was then changed to EFTEM mode and the low-dose module exposure mode was set to nanoprobe mode with a 50 µm C2 aperture at ×215,000 magnification (EFTEM mode). The K2 detector was gain-corrected and the energy filter was fully tuned at the exposure condition. We have updated the previous version of AutoEMation so that it can perform fully automatic VPP data collection, as well as VPP position changes and initial phase-shift buildup every ~40 images, as previously established[23]. During the data collection period, the objective lens was set at eucentric-focus and the specimen was adjusted and imaged at a Z-position of −0.8 µm from the eucentric height for all the exposure holes within an 8 µm radius area. Thirty-two-frame super-resolution movies were collected in a 2.56 s exposure time with a total dose of 50 $e^-$ Å$^{-2}$ and pixel size of 0.26325 Å at the specimen level. Using this method, the data collection speed was ~ 80 images per hour. In total, we collected 1450 movie stacks for apo-SA in a 1-day session and 3309 movie stacks for biotin-SA in a 2-day session.

We used SerialEM to collect VPP electron tomographic data on exactly the same apo-SA cryo-specimen as used in the SPA data collection. Tilt series were collected from −54° to 54° with a 3° interval at ×64,000 magnification (EFTEM mode, pixel size 1.772 Å at the specimen level). For each tilt, the exposure time is 1.0 s with 8 frames using a total dose of 3.38 $e^-$ Å$^{-2}$ in super-resolution mode; therefore, each set of tilt series has a total dose of 125 $e^-$ Å$^{-2}$.

**Image processing**. The super-resolution raw frames of the K2 camera were integrated to MRC format stacks by a local-written program Dat2MRC (developed by Bo Shen, unpublished). MotionCorr (-bin 2 -fod 4 -bft 200 -ssr 1 -ssc 1 -pbx 192) was first used for the full-frame alignment and generated bin2-movie stacks for the initial examination[4]. After the initial examination of the movie stacks for good CTF and astigmatism, the good uncorrected bin2-movie stacks were further processed by MotionCorr2 for a 5 × 5 patch drift correction with dose weighting (-PixSize 0.5265 -kV 300 -Iter 30 -Patch 5 5 -FmDose 1.56 -Bft 200 -Group 3)[50]. The summed bin4-images were generated with a pixel size of 1.053 Å after the MotionCorr2 correction. The non-dose-weighted images were used for the CTF estimation of the defocus, astigmatism, and phase-shift parameters by Gctf[17]. The CTF fitting of each micrograph was examined and screened by checking the Thon ring fitting accuracy manually. The dose-weighted images were used for particle picking and reconstruction. For the apo-SA dataset, 709,967 particles were automatically picked by Gautomatch (developed by Kai Zhang, http://www.mrc-lmb.cam.ac.uk/kzhang/Gautomatch/) from 1385 micrographs. For the biotin-SA dataset, 1,346,980 particles were picked by Gautomatch from 3272 micrographs. After the particles were extracted by Relion, a 120 Å high-pass filter was applied to the particle stacks by relion_image_handler for a better 2D classification performance. The initial model was generated de novo by the 3D initial model in Relion using the SGD method. For each dataset, multiple rounds of 2D or 3D classification were performed in Relion to screen the best particles producing the two 3D reconstructions with a D2 symmetry of apo-SA at about 3.3 Å resolution (23,991 particles) and 3.2 Å resolution (45,686 particles) based on the gold-standard fourier shell correlation (FSC) criterion. In addition, a 3.1 Å resolution reconstruction could be generated by combining the two datasets together with the final refined particles (69,677 particles in total). The local resolution was estimated by the program blocres (Bsoft package)[51]. Directional FSC profiles were calculated on the Remote 3DFSC Processing Server (https://3dfsc.salk.edu/)[52].

For the asymmetric SPA, the star files of the related particles were extended four times by the program relion_particle_symmetry_expand with D2 symmetry. Then, the new star files were input into Relion for the skip-align 3D classification and the particle subtraction following the standard process. Ligand occupancies were calculated by counting valid voxel numbers within a masked biotin region from normalized maps using USCF-Chimera. Three individual 3D classification results were used for estimating the mean value and SD.

To generate the subtracted datasets, the densities to be subtracted were manually adjusted in UCSF-Chimera[53]. The subtracted particles were then re-refined with either local angular search (within 1.8°) or global search from scratch (initial 7.5°). For the global search refinement, the initial models were generated from the target apo-SA maps with 20 Å low-pass filtering.

For the tomography reconstruction, the tilt series raw stacks were first drift-corrected by MotionCor2. The fiducial-free alignment and tomogram reconstruction were done by IMOD's standard procedure[54]. The final tomograms were generated with an eight-time binning (pixel size 7.088 Å) from super-resolution images.

**Model fitting and refinement**. The atomic model of biotin-SA (PDB 1MEP) was fit into the EM density maps as a rigid body in UCSF-Chimera. The crystal structure fit well in the high-resolution EM density maps. Based on the map densities, we mutated and refined some side chains manually in Coot[55] and run one round of real space refinement in PHENIX[56].

**Reporting summary**. Further information on research design is available in the Nature Research Reporting Summary linked to this article.

## Data availability
Data supporting the findings of this manuscript are available from the corresponding authors upon reasonable request. A reporting summary for this Article is available as a Supplementary Information file. The source data underlying Figs. 4c, 6e–g and Supplementary Figs 6A, B, D, 8 A–D are provided as a Source Data file. The accession numbers for the EM maps, models, and raw movie stack of streptavidin reported in this paper are EMD-0689, EMD-0690, PDB-6J6J, PDB-6J6K, EMPIAR-10269, and EMPIAR-10270.

## Code availability
Program Dat2MRC is available from https://github.com/sailorsb/dat2mrc.

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

## Acknowledgements

We thank Xiaomin Li and Tao Yang at the Tsinghua University Branch of the National Protein Science Facility (Beijing) for their technical support on the Cryo-EM and High-Performance Computation platforms. We thank Zhipu Luo at Soochow University for his help in atomic model refinement. This work was supported by grant (2016YFA0501100 to H.W. and J.L., 2016YFA0200101 to H.P.) from the Ministry of Science and Technology of China, grant (Z161100000116034 to H.W.) from the Beijing Municipal Science & Technology Commission, and grant (21525310 to H.P.) from the National Natural Science Foundation of China.

## Author contributions

X.F., J.W., J.L., and H.W. conceived the experiments. X.F. and H.W. wrote/corrected the manuscript. X.F. performed the cryo-EM sample preparation and data collection. X.F. and J.W. performed SPA data processing. X.Z. and X.F. performed the cryo-ET analysis. X.F. and Z.Y. performed the statistical analysis of particle distribution. J.Z. and H.P. prepared graphene grids. L.Z. and J.L. aided in hardware setup and data collection.

## Additional information

**Competing interests:** The authors declare no competing interests.

