## [Peer Review File · Nature Communications]

Reviewers' Comments:

Reviewer #1:

Remarks to the Author:

Xiao et al describe the use of cryo-EM single particle analysis (SPA) with both Volta phase plate (VPP) and spherical aberration (CS) correction technology to reconstruct the 52 kDa Streptavidin protein to 3.2 Å resolution. They extend some aspects of the workflow to a 39 kDa Streptavidin trimer. The manuscript provides interesting updates to prior work by Khoshouei, M., Radjainia, M., Baumeister, W., & Danev, R. (2017). *Nature Communications*, 8, 16099, wherein the authors reported a 3.2 Å resolution reconstruction of a 64 kDa particle using a similar methodology. In addition, the current work begins to address several important questions for the cryo-EM field, such as the application of graphene for single-particle imaging, practical size limitations and particle quality for high-resolution reconstructions, the effects of adsorption to non-solvent interfaces, and the combination of the cryo-EM with cryo-ET to investigate the behavior of single-particle samples. All of these developments contribute useful insight into particle behavior on cryo-EM grids, although they are not necessarily novel per se. In this reviewer's opinion, the manuscript could, eventually, be ready for publication in *Nature Communications*. However, the scope of the remaining work, specifically to justify and properly control all their conclusions, is significant. As is, I cannot recommend it for publication. I will elaborate below.

Major points

The authors touch upon numerous interesting, and important points that are relevant to the cryo-EM field. However, none of the individual points is adequately explored, controlled, or brought to definitive conclusions. I will highlight some of the putative advances and the major remaining questions

(1) The authors applied SPA methods to solve the smallest protein structure to date at near-atomic resolution. However, what was the trick? Was it the graphene? Was it the VPP? Was it the Cs corrector? Was it the size of the dataset? Should we all be purchasing Titan Krioses with Cs correctors and VPPs? We now know that small proteins can also be solved using conventional methods (Herzik, M. A., Jr., Wu, M., & Lander, G. C. (2018). High-resolution structure determination of sub-100 kilodalton complexes using conventional cryo-EM. *bioRxiv*, 489898. <http://doi.org/10.1101/489898>). What is the advantage of the proposed workflow?

(2) The authors suggest that small proteins, less than 39 kDa, could potentially be solved using cryo-EM methods. Based on the experiments performed, it is not clear under what conditions the current approaches begin to fail. Density subtraction is a clever approach for evaluating the potential, but of course, it is imperfect. There will be residual density (as is evident in the class averages in Figure 5B), and the sample will generally behave differently from a bona fide, biochemically purified monomer/dimer/trimer (e.g. grid preparation will change, orientation distribution will change, etc.). Importantly, it is critical to show the actual reconstructions from global refinement (Figure 5C). Check marks and crosses don't qualify as data. The reader needs to see what the reconstructions look like, what is the resolution, orientation distribution, where are the problems in the maps, etc. Thus, it is not currently clear what we can conclude from this experiment, and to what extent are these results actually translatable to a real dataset?

(3) The use of graphene as a substrate support for imaging small proteins. This, as the authors point out, appears to be one of the first applications of graphene technology to the analysis of small proteins (graphene has been used before in numerous papers). A very important consideration for the field is: how reproducible, and how easy to make, are graphene grids for SPA? I think it's necessary to elaborate on this, at least as supplemental material. One important observation from this paper, which also highlights the potential limitation of graphene, is the clustering of particles on the GWI. Presumably, this also implies that there is preferred specimen orientation and anisotropy. What is the orientation distribution of the particles? How does glow

discharging affect particle adsorption and orientation distribution? (the authors commented about glow discharging leading to putative aggregation). How do these affect resolution? The ability to perform global reconstructions?

(4) Building upon the above point, it is still not clear to me what graphene is actually doing, and whether it is contributing to the improvements in resolution. To conclusively test this, one would need to have a dataset without graphene. At a minimum, one would want to see particles within the current dataset separated between those lying on the GWI and those on the AWI. I understand that the two cannot be perfectly deconvoluted without collecting a tomogram for every imaged area, but one can, at least in part, infer this information from the particle location. How do the orientation distributions, resolutions, etc. differ between the two? If this is not possible, its necessary to collect a control dataset without graphene.

(5) The authors suggest the use of focused classification to understand ligand binding. This is a good idea, and is something that would have implications for looking at drug-bound complexes. Of course, focused classification is nothing new, and neither is looking at ligand-bound complexes. The authors' claim here is that the combination of the two is novel, rather than any individual step. But then, I would argue that much more should be done. Can the small molecule (biotin) be modeled? With what accuracy? What is the occupancy of each asymmetric unit? How does the density (and the model) of biotin differ from one asymmetric unit to another? In other words, what is the real applicability of focused classification for detecting small molecules? Is it a real useful tool, or just something that one might want to try? Furthermore, there is always the possibility that an empty class results from bad particles that were not discarded during classification, and not from sub-stoichiometric occupancy of the inhibitor. This needs to be controlled. As is, I cannot currently draw any meaningful conclusion with respect to small molecule binding from Figure 4.

(6) The use of tomography to understand the spatial distribution of particles on a grid. This is important, and has been recently described in: Noble, A. J., Dandey, V. P., Wei, H., Brasch, J., Chase, J., Acharya, P., et al. (2018). Routine single particle CryoEM sample and grid characterization by tomography. *eLife*, 7, 32.

The rationale that the improvement from 64 kDa to 52 kDa (and possibly to 39 kDa, although there is no experimental density for a de novo structure of trimeric SA) is a major advance seems to predicate on the assumption that SA consists predominantly of beta-strands, rather than alpha helices, which, as the authors argue, compromises the ability to properly align the particles. But is this really true? What is the evidence for this? The authors need to justify this with an experiment or point to relevant literature that thoroughly explores this phenomenon.

In the abstract: "We demonstrated that the method is capable to determine the structure of molecule as small as 39 kDa and potentially even smaller molecules." The results in the manuscript do not support the above statement. Specifically, the authors' experiments with signal-subtracted (note: not biochemically purified) particles suggested that ab initio reconstructions were not possible for particles smaller than 39kDa.

In the abstract: "Furthermore, we found that using the graphene film to avoid the adsorption to the air-water interface is critical to maintain the protein's high-resolution structural information". As mentioned above, this claim is underdeveloped. At a minimum, a thorough investigation of how different particles either lying at the GWI or the AWI contribute to the reconstruction, within the current dataset, needs to be conducted. Alternatively, a control dataset without graphene should be collected and processed in a comparable manner.

The authors need a table S1 to show all map refinement statistics. Ideally, and building on the points above, this should also include a model and contain modeling statistics.

FSC curves, local resolution maps, Euler plots, and anisotropy profiles should be displayed for all

relevant structures discussed in the text.

What is happening to the 80% of the particles that are discarded? Are they at the AWI? Are they at the GWI, but aggregating? Are they anisotropic? Again, a thorough investigation needs to be conducted.

Minor points

Why is 200 kDa "challenging"? Why not 100 kDa? Why not 50 kDa? Presumably, there is no one sharp cut-off.

Figure 4C and Figure S5. A map of local resolution should be included with a highlight of the biotin-binding pocket. Also, what is the percentage of particles within each class?

Figure 5B. Please align the class averages to one another for easier comparison.

Figure S3. Please show the results of classification. Which volumes were discarded? Why?

Page 8: I believe the authors meant 4-times larger, rather than 3-times larger (the particle count was multiplied by 4 by expanding the symmetry from D2, even though the number of additional, "new" copies of asymmetric units was 3)

There are many grammatical errors throughout the manuscript, and it would help to have a careful proof-read prior to resubmission. Some examples, from the very beginning of the paper, are below:

Abstract: "high-resolution structures" instead of "high resolution structure"

Page 3: "technologies" instead of "technology"

Page 3: "in recording speed" instead of "on the recording speed"

Page 3: "restored" is probably not the right term. "recovered"?

Page 3: "to reconstruct" instead of "to get the reconstruction of a ..."

Page 4: remove "the" from "the frozen-hydrated ..."

Page 4: "it was shown" instead of "it has been proved" (or consider revising the sentence altogether)

... etc.

Reviewer #2:

Remarks to the Author:

In this article, the authors present the smallest to date protein structure solved at near-atomic resolution by cryo-EM, that of streptavidin. They convincingly demonstrate the potential of cryo-EM to solve the structures of even smaller molecules and to detect the presence of a small ligand. Furthermore, they explore the effects of the air-water interface on the quality of the sample and the benefits of using a thin graphene layer as specimen support. The paper represents a significant methodological advance and would be of great interest to the audience of Nature Communications. It will have a substantial impact in the fields of structural biology and drug discovery. Therefore, I strongly recommend its publication, pending minor revisions.

Suggested revisions:

1. The grammar and style of the manuscript need improvement.
2. The authors use the term "well behaved molecules/proteins" in both the abstract and the introduction but do not provide an explanation for it. Please provide a short description e.g. "well-behaved proteins, in terms of homogeneity, rigidity and random orientations in ice." for readers who are not familiar with the requirements of cryo-EM.
3. It is surprising that less than 5% of picked particles reached the final 3D reconstructions. The

bimodal distribution of particles on the graphene and air-water interfaces only partially explains this result. Did the authors analyze the relation between particle retention and ice thickness, as judged by the average image intensity? If there is a correlation, please include this information in the paper.

4. The higher density of particles on the graphene surface and the graphene layer itself may have provided a stronger signal and therefore may have biased the CTF parameter fits. This could be an alternative explanation for their predominant retention in the final high-resolution subsets.

5. The molecules on the air-water interface in Fig. 6C look subjectively better (have higher contrast) than the molecules on the graphene surface. Do the authors have an explanation for this difference in contrast?

6. The VPP-Cs-corrector combination is mentioned several times throughout the manuscript. The possible contribution of the VPP to the success of the investigation is the improved image contrast. However, it is unclear what the potential contribution of the Cs-corrector was. Please include a comment about the possible benefits from using an image corrector.

Radostin Danev

Reviewer #3:

Remarks to the Author:

The authors report near-atomic resolution structures of streptavidin and biotin-bound streptavidin obtained by cryoelectron microscopy. They demonstrate that the ligand-bound state can be classified out from the composite dataset and that a molecule as small as the trimeric streptavidin can potentially be reconstructed to high resolution without prior knowledge. These findings demonstrate that proteins with molecular weight of 50 kDa or even smaller are tractable by the current single-particle cryo-EM technology. The authors used a TEM equipped with a Volta phase plate and a Cs corrector; the relative importance of these for the success of this work requires further discussion.

A graphene support film was used in preparing the vitrified specimen for these studies. However, the graphene seems not properly wetted, probably due to its hydrophobicity and the insufficient effect of the glow discharge treatment. This can lead to artefacts such as clustered particles, variations in background signal, dried out particles with affected structural integrity, which the authors attempt to deal with during the processing of the data and misinterpret as problems due to interactions at the air-water interface.

Major remarks

1. The authors speculate about the better structural feature preservation in particles from 'clustering areas' (page 13) and suggest that these areas might be formed due to uneven glow-discharge on the surface.

The authors must provide a physical reasoning regarding what makes the plasma uneven at the described sub-micron scale. Another potential reason, however, is that the protein solution is partially dried out on the supporting film with only the clustered particles embedded in vitreous ice. For example, the micrographs in Figure 6 and Supplementary Figure 1 look like the graphene is only partially wetted and some regions are dried out. This non-uniform wetting may occur if the graphene surface is not sufficiently hydrophilic due to poor glow discharge or is severely contaminated with other species. The non-clustered protein particles may then be dried onto the surface without hydration and therefore have their structure disrupted, while in the cluster some amount of ice is preserved. The authors must show that this is not the case. From the tomographic cross-sections shown in Supplementary Figure 6 it is not obvious that the clusters localize to the graphene surface and the remaining particles to the air-water interface. If the authors cannot provide direct evidence, the speculation that the SA particles attached to the graphene are better

structurally preserved (page 14) must be moved to the Discussion section or removed completely.

2. The authors mention the use of a Cs corrector in this work. They should comment why they think spherical aberrations are a limiting factor in cryo-EM of small proteins and why a Cs corrector was necessary. Otherwise, the authors must make it clear that the Cs corrector is not a central part of this experiment, i.e. the same results are achievable without one. Failure to clarify this point may lead the field to misconceptions regarding the necessity of Cs-corrected TEMs for cryo-EM.

3. The authors state that single-crystalline monolayer graphene was used in this study (page 5) but no data is provided to demonstrate that the graphene was indeed a single crystal and monolayer. The authors should experimentally characterize the graphene to support this statement (eg. as in Meyer et al, Nature 2017) or remove it from the text.

4. In the discussion section (page 15) the authors attempt to list the potential benefits of using graphene support films. However, there is no evidence to support statement (1) that the graphene reduces ice thickness, and (2) that the graphene reduces charging, radiation damage, and motion. The authors must provide supporting evidence for these claims or remove them. While the tomograms in Supplementary Figure 6 do indeed show that ice thickness of 200 – 500Å was achieved, this is also possible without the use of supporting films. To demonstrate reduced radiation damage and motion, the authors must provide B-factor plots and motion statistics. It is highly recommended to provide these in any case when reporting a cryo-EM structure.

5. It is also highly recommended that the authors provide the orientation distributions of the particles on the graphene, as calculated via the 3D refinement. The authors may also wish to comment whether they attempted at all to solve these structures without the use of a supporting graphene film, as this will help elucidate the importance of the graphene support in this work.

6. The authors conclude the section 'Focused classification analysis of the biotin binding pocket of SA' with the statement 'The results above proved the capability of heterogeneity analysis for ligand binding detection of small proteins by single particle cryo-EM'. However, the specific case demonstrated here incorporated roughly equal number of particles with and without a ligand bound, which may not be the case in general. The authors may investigate this further (eg. whether a minority ligand-bound class can be separated out) or alternatively edit/remove the above generalizing statement. The same applies to the related statement in the Discussion (page 15, beginning 'One may solve the structures...') since the authors do not demonstrate in this work that different ligand-occupied states can be classified. The authors should also cite other recent studies where ligand-bound and unbound states of proteins were separated by cryo-EM.

7. Further in the discussion section (page 15), the authors say that it was unexpected to find the particles in the specimen localize to the interfaces. However, this has been known to be the case since the very early days of cryo-EM (Dubochet, 1985) and more recent tomographic studies demonstrate the phenomenon clearly (Noble et al, 2018). This statement should be edited accordingly (eg. 'In agreement with other studies we found that the SA particles localize to the interfaces...') and the following references should be added: Noble et al, eLife (2018) and Dubochet et al, Trends in Biochem Sci (1985).

8. The authors also claim that the problem of absorption to the air-water interface is more severe for smaller proteins (page 3-4) because the ice is thinner, which is incorrect, so this statement should be removed.

Minor remarks

9. Any claims regarding the smallest protein structure solved using cryo-EM with the aid of certain methods (eg. phase plate, supporting films) should be carefully revised to match the current state

of the field. The authors should consider citing Herzik et al (bioRxiv, 2018) who report high-resolution structures of 82 kDa and 64 kDa proteins by cryo-EM without the use of support films, a phase plate or aberration correctors. The authors should also cite Herzik et al (Nature Methods, 2017) for the structure determination of the 150 kDa enzyme aldolase using a TEM operated at 200 kV.

10. The authors mention that high-pass filtering the particles (~50 Å in size) to 120 Å turned out to be necessary for their alignment and show that this improved the 2D classes (Supplementary Figure 2). The relevance of this filtering with frequency more than twice bigger than the particle size requires clarification. Why does it improve the alignment? For example, was this used to remove uneven background due to the supporting graphene film? This is likely to be the case if the graphene surface was not properly wetted as mentioned above, see eg. Fig. 23 in Dubochet et al, 1988.

11. The authors should clarify the duration of the glow-discharge treatment of the graphene-coated grids (page 17). In the same section, the authors may like to clarify why the blotting settings were different for the apo-SA and biotin-bound SA samples, given that the same grids and same concentration samples were used.

12. The authors say that particles reach the interfaces 'quite fast' (page 16); they may refer to Glaeser (Curr Opin in Colloid Interface Sci, 2018) and the Supplement of Naydenova & Russo (Nature Comm, 2017), for a quantitative estimate of exactly how fast this occurs.

13. Acronyms for the air-water interface (AWI) and the graphene-water interface (GWI) are not widely used and only make the text less readable, therefore it is recommended that these are removed.

14. The manuscript requires major corrections for spelling and grammar, which can significantly improve the readability of the text. For example, on page 17 'plugged' should read 'plunged', etc.

Our response to the reviewers' comments are in **bold font**.

Reviewers' comments:

Reviewer #1 (Remarks to the Author):

Xiao et al describe the use of cryo-EM single particle analysis (SPA) with both Volta phase plate (VPP) and spherical aberration (CS) correction technology to reconstruct the 52 kDa Streptavidin protein to 3.2 Å resolution. They extend some aspects of the workflow to a 39 kDa Streptavidin trimer. The manuscript provides interesting updates to prior work by Khoshouei, M., Radjainia, M., Baumeister, W., & Danev, R. (2017). *Nature Communications*, 8, 16099, wherein the authors reported a 3.2 Å resolution reconstruction of a 64 kDa particle using a similar methodology. In addition, the current work begins to address several important questions for the cryo-EM field, such as the application of graphene for single-particle imaging, practical size limitations and particle quality for high-resolution reconstructions, the effects of adsorption to non-solvent interfaces, and the combination of the cryo-EM with cryo-ET to investigate the behavior of single-particle samples. All of these

developments contribute useful insight into particle behavior on cryo-EM grids, although they are not necessarily novel per se. In this reviewer's opinion, the manuscript could, eventually, be ready for publication in *Nature Communications*. However, the scope of the remaining work, specifically to justify and properly control all their conclusions, is significant. As is, I cannot recommend it for publication. I will elaborate below.

Major points

The authors touch upon numerous interesting, and important points that are relevant to the cryo-EM field. However, none of the individual points is adequately explored, controlled, or brought to definitive conclusions. I will highlight some of the putative advances and the major remaining questions

(1) The authors applied SPA methods to solve the smallest protein structure to date at near-atomic resolution. However, what was the trick? Was it the graphene? Was it the VPP? Was it the Cs corrector? Was it the size of the dataset? Should we all be purchasing Titan Krios with Cs correctors and VPPs? We now know that small proteins can also be solved using conventional methods (Herzik, M. A., Jr., Wu, M., & Lander, G. C. (2018). High-resolution structure determination of sub-100 kilodalton complexes using conventional cryo-EM. *bioRxiv*, 489898. <http://doi.org/10.1101/489898>). What is the advantage of the proposed workflow?

Thanks for the reviewer's comments. We all know that the major hurdle that all cryo-EM practitioners are trying to overcome is the low signal to noise ratio (SNR) at both low frequency and high frequency. The contrast related to the low-frequency SNR is more critical for studies of small proteins. There are multiple ways to improve the contrast, including defocus contrast, low-voltage TEM, applying phase-plate, higher DQE of the camera and reduce ice thickness at sample level. If the low frequency signal is enough for a 3D-refinement

and the micrograph contains enough high resolution information, one should succeed. We never claim that VPP is the only way for small protein analysis and we truly appreciate the work done by Herzik, M. A., Jr., Wu, M., & Lander, G. C. (2018). Instead, capturing signal as much as possible using all available methods is probably the best practice to push the boundary of cryo-EM as far as possible. The application of VPP is one of the most efficient ways. Neither do we claim that Cs-corrector is the only way to solve high resolution structures. But we did demonstrate that the usage of Cs-corrector in combination with VPP could allow new imaging strategies without worrying over- or under-focus during the data acquisition [Fan X, Zhao L, Liu C, et al. Near-Atomic Resolution Structure Determination in Over-Focus with Volta Phase Plate by Cs-Corrected Cryo-EM.[J]. *Structure*, 2017, 25(10):1623.]. Such a strategy allowed us to collect high-quality dataset more efficiently.

As for the usage of the graphene film as supporting material, we found it is very useful in our study of streptavidin. We now have elaborated more on the destructive effect of air-water interface on the streptavidin molecule in our new analysis of the dataset. Our new results demonstrated clearly that graphene was critical for us to solve the structure of streptavidin at 3.2 Angstrom resolution. We have also included more detail explanation of the preparation of the graphene grid in the Materials and Methods of the manuscript.

The size of the dataset for streptavidin is relatively small in comparison with many other single particle datasets in EMDB. This may be partially due to the application of VPP and graphene.

As for the question: “Should we all be purchasing Titan Krioses with Cs correctors and VPPs?”, we do not have any intention to make such kind of claim. We simply demonstrated what we obtained using such a setup. We now added additional discussion about the Cs-corrector’s role in the revision. We would be happy to test our specimen on other instrument setups and cryo-EM strategies such as the beautifully demonstrated examples by the Lander group at the Scripps Research Institute in the future.

Overall, our current work does not intend to provide a one-time solution for all cryo-EM technical problems. Neither did the previous work by Khoshouei et al. on the 64 kDa hemoglobin nor the recent work by Herzik et al. using conventional cryo-EM. We are happy that our efforts help push the limit of cryo-EM in a useful way.

(2) The authors suggest that small proteins, less than 39 kDa, could potentially be solved using cryo-EM methods. Based on the experiments performed, it is not clear under what conditions the current approaches begin to fail. Density subtraction is a clever approach for evaluating the potential, but of course, it is imperfect. There will be residual density (as is evident in the class averages in Figure 5B), and the sample will generally behave differently from a bona fide, biochemically purified monomer/dimer/trimer (e.g. grid preparation will change, orientation distribution will change, etc.). Importantly, it is critical to show the actual reconstructions from global refinement (Figure 5C). Check marks and crosses don’t qualify as data. The reader needs to see what the reconstructions look like, what is the resolution, orientation distribution, where are the problems in the maps, etc. Thus, it is not currently clear what we can conclude from this experiment, and to what extent are

these results actually translatable to a real dataset?

Thanks for the reviewer's comments and suggestions. Firstly, we are sorry for the non-informative labels in Figure 5C. We have modified Figure 5C as suggested by the reviewers to include the reconstructions of monomer and dimer from the global search.

Secondly, in this experiment/simulation we want to address two questions. 1). If we have a prior knowledge of the angular information of each particle, could we successfully extract the 3D structure information from a very small protein (i.e. 13 kDa) using nowadays software? The answer is yes: the reconstruction process succeeded from local angular search refinement given roughly the correct angular information of each particle. 2). How small molecules could be potentially well aligned from scratch with a similar data quality (SNR)? From the 4 dataset classification and refinement results, we observed the following tendency: 26 kDa dataset could give a precise 2D alignment in at least one view and 39 kDa dataset could give the correct 3D refinement from global search. This tendency consists with the increase of low frequency signals that facilitate the image alignment. This led us to infer that data collected using our approach had the potential to reconstruct the correct structure of protein of ~39 kDa. We have now revised our manuscript to explain the results more accurately.

We totally agree with the reviewer that a real dataset with molecular weight below 40 kDa would be worth further investigating. This could be our next goal to study smaller proteins to further push the lower limit of molecular weight by single particle cryo-EM.

(3) The use of graphene as a substrate support for imaging small proteins. This, as the authors point out, appears to be one of the first applications of graphene technology to the analysis of small proteins (graphene has been used before in numerous papers). A very important consideration for the field is: how reproducible, and how easy to make, are graphene grids for SPA? I think it's necessary to elaborate on this, at least as supplemental material. One important observation from this paper, which also highlights the potential limitation of graphene, is the clustering of particles on the GWI. Presumably, this also implies that there is preferred specimen orientation and anisotropy. What is the orientation distribution of the particles? How does glow discharging affect particle adsorption and orientation distribution? (the authors commented about glow discharging leading to putative aggregation). How do these affect resolution? The ability to perform global reconstructions?

Thanks for the reviewer's comments. We are sorry that the usage of graphene film was not explained clear enough in our manuscript. We (co-author Hai-Lin Peng) have previously published a method to prepare and characterize the high quality single crystalline graphene-coated EM grids [Zhang, J. et al. Single Crystals: Clean Transfer of Large Graphene Single Crystals for High-Intactness Suspended Membranes and Liquid Cells, *Advanced Materials* 29 (2017)]. We were using the very similar protocols that was described in the paper. We now provide more detail about the grid preparation in the Materials and Methods of the revision.

As for the clustering phenomenon, we used rather mild glow-discharge (10 second at low level in a Harrick Plasma instrument) on the graphene grids. This likely caused partial hydrophilicity on the surface. Following the reviewers' suggestions, we have performed more careful analysis of particles on GWI and AWI to understand the effect of different surfaces on the particle's orientation and structure. We did observe a major difference between particles on GWI and AWI. The particles on GWI are better preserved and have more orientations than those on AWI. The particles on AWI suffered from preferential orientation and failed to reconstruct to high resolution. These results are now included in the revised manuscript.

(4) Building upon the above point, it is still not clear to me what graphene is actually doing, and whether it is contributing to the improvements in resolution. To conclusively test this, one would need to have a dataset without graphene. At a minimum, one would want to see particles within the current dataset separated between those lying on the GWI and those on the AWI. I understand that the two cannot be perfectly deconvoluted without collecting a tomogram for every imaged area, but one can, at least in part, infer this information from the particle location. How do the orientation distributions, resolutions, etc. differ between the two? If this is not possible, its necessary to collect a control dataset without graphene.

This is an excellent point. At least in the case of streptavidin, we failed to make good cryo-EM specimen of the protein suspended in thin ice, therefore graphene film seemed critical for the specimen preparation. We followed the reviewer's suggestion to separate and process particles from 749 micrographs that have clear clustering features into two subsets, among which the subset A contains most of the particles on GWI and the subset B comprises mainly of particles on AWI. We analyzed and processed the two datasets separately and did find difference between them regarding the orientation distribution, particle retention and reconstruction quality. These new results are now included in our revision and we are happy that they further strengthen the conclusion. Thanks to the reviewer!

(5) The authors suggest the use of focused classification to understand ligand binding. This is a good idea, and is something that would have implications for looking at drug-bound complexes. Of course, focused classification is nothing new, and neither is looking at ligand-bound complexes. The authors' claim here is that the combination of the two is novel, rather than any individual step. But then, I would argue that much more should be done. Can the small molecule (biotin) be modeled? With what accuracy? What is the occupancy of each asymmetric unit? How does the density (and the model) of biotin differ from one asymmetric unit to another? In other words, what is the real applicability of focused classification for detecting small molecules? Is it a real useful tool, or just something that one might want to try? Furthermore, there is always the possibility that an empty class results from bad particles that were not discarded during classification, and not from sub-stoichiometric occupancy of the inhibitor. This needs to be controlled. As is, I cannot currently draw any meaningful conclusion with respect to small molecule binding from Figure 4.

Thanks for the reviewer's comments. At current resolution $\sim 3.2 \text{ \AA}$ of our reconstruction, we can identify the location of the ligand but may not be able to build its atomic model without prior knowledge.

We used the mixed dataset to mimic a situation of partial occupancy of biotin ligand. As demonstrated in Figure 4 A and B of our manuscript, the reconstruction from the mixed dataset showed a density corresponding to the biotin ligand. Such a ligand occupancy became clearer after 3D classification. As a control experiment, the dataset of apo-SA or biotin-SA only yielded reconstructions without biotin density or with biotin density after classification, respectively (Figure S5). Although the current resolution is not enough to unambiguously build atomic model of the ligand, we would argue that localization of a ligand binding pocket is also very important for drug discovery.

To further understand the effectiveness of classification in identifying ligand binding pocket, we generated multiple mixed datasets of asymmetric particles to mimic various ligand occupancy conditions and performed 3D reconstructions of them. We demonstrated that the classification could separate the ligand-binding state from the apo state to reveal the location of the ligand. We have updated our manuscript with the new analysis results.

(6) The use of tomography to understand the spatial distribution of particles on a grid. This is important, and has been recently described in: Noble, A. J., Dandey, V. P., Wei, H., Brasch, J., Chase, J., Acharya, P., et al. (2018). Routine single particle CryoEM sample and grid characterization by tomography. *eLife*, 7, 32.

The rationale that the improvement from 64 kDa to 52 kDa (and possibly to 39 kDa, although there is no experimental density for a de novo structure of trimeric SA) is a major advance seems to predicate on the assumption that SA consists predominantly of beta-strands, rather than alpha helices, which, as the authors argue, compromises the ability to properly align the particles. But is this really true? What is the evidence for this? The authors need to justify this with an experiment or point to relevant literature that thoroughly explores this phenomenon.

Thanks for the reviewer to point this out. We currently don't have solid evidence if beta-sheet containing molecules are more difficult to align than the alpha-helix containing molecules. This should be further investigated. We have deleted this statement in the revision.

In the abstract: "We demonstrated that the method is capable to determine the structure of molecule as small as 39 kDa and potentially even smaller molecules." The results in the manuscript do not support the above statement. Specifically, the authors' experiments with signal-subtracted (note: not biochemically purified) particles suggested that ab initio reconstructions were not possible for particles smaller than 39kDa.

We have toned down our claim by deleting the "and potentially even smaller molecules" from the abstract.

In the abstract: “Furthermore, we found that using the graphene film to avoid the adsorption to the air-water interface is critical to maintain the protein’s high-resolution structural information”. As mentioned above, this claim is underdeveloped. At a minimum, a thorough investigation of how different particles either lying at the GWI or the AWI contribute to the reconstruction, within the current dataset, needs to be conducted. Alternatively, a control dataset without graphene should be collected and processed in a comparable manner.

As we have shown by the new analysis in the revision, the particles on GWI are better preserved than those on AWI. We have changed the sentence in the abstract to a new sentence as: “Furthermore, we found that avoiding the adsorption of proteins to the air-water interface is critical to maintain the high-resolution structural information.”

The authors need a table S1 to show all map refinement statistics. Ideally, and building on the points above, this should also include a model and contain modeling statistics.

We have included a Table S1 for cryo-EM and model statistics now.

FSC curves, local resolution maps, Euler plots, and anisotropy profiles should be displayed for all relevant structures discussed in the text.

We have provided the necessary materials as suggested.

What is happening to the 80% of the particles that are discarded? Are they at the AWI? Are they at the GWI, but aggregating? Are they anisotropic? Again, a thorough investigation needs to be conducted.

We were also eager to understand the situation of the 80% particles that were discarded during the refinement. That’s why we performed the cryo-ET analysis of the specimen and discovered that particles distributed on two different interfaces. From our new analysis of the particles’ distribution on the two interfaces and their performance in 2D and 3D classification, we may conclude that a major portion of the bad particles were on the AWI. There may also be certain portion of particles with conformational heterogeneity that exclude them from the high-resolution class.

Minor points

Why is 200 kDa “challenging”? Why not 100 kDa? Why not 50 kDa? Presumably, there is no one sharp cut-off.

Based on the statistical analysis on released maps in EMDB, more than 2,000 released maps obtained at the resolution higher than 5Å. Among these, no more than 20 proteins have molecular weight smaller than 200 kDa, and more than half of them are membrane proteins whose detergent micelle increases the contrast of the particles. We agree with the reviewer that there’s no sharp cut-off. But with less than 1% statistical success rate for high-resolution reconstruction, we feel that 200 kD is a reasonable value as “challenging” for the current status.

Figure 4C and Figure S5. A map of local resolution should be included with a highlight of the biotin-binding pocket. Also, what is the percentage of particles within each class?

We have included the local resolution map in Figure S4 for the biotin-SA reconstruction and highlighted the biotin-binding pocket in the revision. We also included the percentage of particles within each class.

Figure 5B. Please align the class averages to one another for easier comparison.

We have revised the figure as suggested.

Figure S3. Please show the results of classification. Which volumes were discarded? Why?

We have revised the manuscript to demonstrate the image processing workflow in more detail showing the classification results in Figure S3 and S4.

Page 8: I believe the authors meant 4-times larger, rather than 3-times larger (the particle count was multiplied by 4 by expanding the symmetry from D2, even though the number of additional, “new” copies of asymmetric units was 3)

We have revised the manuscript as suggested.

There are many grammatical errors throughout the manuscript, and it would help to have a careful proof-read prior to resubmission. Some examples, from the very beginning of the paper, are below:

Abstract: “high-resolution structures” instead of “high resolution structure”

Page 3: “technologies” instead of “technology”

Page 3: “in recording speed” instead of “on the recording speed”

Page 3: “restored” is probably not the right term. “recovered”?

Page 3: “to reconstruct” instead of “to get the reconstruction of a ...”

Page 4: remove “the” from “the frozen-hydrated ...”

Page 4: “it was shown” instead of “it has been proved” (or consider revising the sentence altogether)

... etc.

We have revised the manuscript carefully for the language.

Reviewer #2 (Remarks to the Author):

In this article, the authors present the smallest to date protein structure solved at near-atomic resolution by cryo-EM, that of streptavidin. They convincingly demonstrate the potential of cryo-EM to solve the structures of even smaller molecules and to detect the presence of a small ligand. Furthermore, they explore the effects of the air-water interface on the quality of the sample and the benefits of using a thin graphene layer as specimen support. The paper represents a significant methodological advance and would be of great interest to the audience of Nature Communications. It will have a substantial impact in the fields of structural biology and drug discovery. Therefore, I strongly recommend its publication, pending minor revisions.

Suggested revisions:

1. The grammar and style of the manuscript need improvement.

We have revised the language in the manuscript carefully.

2. The authors use the term “well behaved molecules/proteins” in both the abstract and the introduction but do not provide an explanation for it. Please provide a short description e.g. “well-behaved proteins, in terms of homogeneity, rigidity and random orientations in ice.” for readers who are not familiar with the requirements of cryo-EM.

We have revised the manuscript as suggested.

3. It is surprising that less than 5% of picked particles reached the final 3D reconstructions. The bimodal distribution of particles on the graphene and air-water interfaces only partially explains this result. Did the authors analyze the relation between particle retention and ice thickness, as judged by the average image intensity? If there is a correlation, please include this information in the paper.

From the tomographic analysis, the general ice thickness of areas for data acquisition is about 10-30 nm. In this small range, we found it difficult to estimate the ice thickness variation by examining the average image intensity because we did not use an energy filter to collect the data. Besides the particles adsorbed to the air-water interface, there may still be a portion of particles with conformational heterogeneity in our samples caused by other factors.

4. The higher density of particles on the graphene surface and the graphene layer itself may have provided a stronger signal and therefore may have biased the CTF parameter fits. This could be an alternative explanation for their predominant retention in the final high-resolution subsets.

Thanks for the reviewer’s comments. It is true that graphene film and a condensed particle distribution would have stronger signal for CTF estimation. If so, this would potentially make the CTF fitting value towards to the graphene plane. However, from the tomography analysis and the area we chose for data collection, the general ice thickness is ~10-30 nm. Without considering any tilt, defocus error at this level (up to 30 nm) would not introduce a 0.5 pi phase error for 500 nm defocus at ~4 Å (as shown below), thus may not influence the rough angular search at first round of 3D classification.

More importantly, our new analysis on two different subsets of particles based on their locations in the clustering region (subset A) and uniformly dispersed region (subset B) has shown that the latter subset (mostly containing particles on AWI) indeed suffered from preferential orientation and less well preserved structures (shown in Figure S8 of our revised manuscript). Thus CTF estimation error plays little role in comparison to the structure preservation for the retention of particles on GWI in the high-resolution reconstruction.

5. The molecules on the air-water interface in Fig. 6C look subjectively better (have higher contrast) than the molecules on the graphene surface. Do the authors have an explanation for this difference in contrast?

We also noticed this phenomenon. We have two possible hypotheses: 1) the molecules on the air-water interface have a strong preferential orientation along its elongated direction and this orientation would give a higher contrast; 2) the molecules on the air-water interface were somehow partially denatured and condensed for their structural elements sticking in the air. Our new results in Figure S8 support these hypotheses.

6. The VPP-Cs-corrector combination is mentioned several times throughout the manuscript. The possible contribution of the VPP to the success of the investigation is the improved image contrast. However, it is unclear what the potential contribution of the Cs-corrector was. Please include a comment about the possible benefits from using an image corrector.

We need to clarify that we were solely stating the fact that our microscopy was performed on a microscope with a VPP-Cs-corrector combination. We have no intention to claim that the Cs-corrector is necessary to get the high-resolution structure of 52 kDa streptavidin. However, the Cs-corrector in combination with VPP did provide us with a convenience to use a relatively more efficient data collection strategy because we do not need to worry the under-focus or

over-focus conditions of the objective lens as described in our previous publication [Fan X, Zhao L, Liu C, et al. Near-Atomic Resolution Structure Determination in Over-Focus with Volta Phase Plate by Cs-Corrected Cryo-EM. *Structure*, 2017, 25(10):1623.]. In brief, we pre-recorded some in-focus Z-heights to speed up data collection (by skip focusing) and improve VPP behavior (by without touching the objective lens). We would like to perform the similar analysis on a microscope equipped with VPP but no Cs-corrector in the future. We have added a paragraph to discuss more on the Cs-corrector's role in the revised manuscript.

Reviewer #3 (Remarks to the Author):

The authors report near-atomic resolution structures of streptavidin and biotin-bound streptavidin obtained by cryoelectron microscopy. They demonstrate that the ligand-bound state can be classified out from the composite dataset and that a molecule as small as the trimeric streptavidin can potentially be reconstructed to high resolution without prior knowledge. These findings demonstrate that proteins with molecular weight of 50 kDa or even smaller are tractable by the current single-particle cryo-EM technology. The authors used a TEM equipped with a Volta phase plate and a Cs corrector; the relative importance of these for the success of this work requires further discussion.

A graphene support film was used in preparing the vitrified specimen for these studies. However, the graphene seems not properly wetted, probably due to its hydrophobicity and the insufficient effect of the glow discharge treatment. This can lead to artefacts such as clustered particles, variations in background signal, dried out particles with affected structural integrity, which the authors attempt to deal with during the processing of the data and misinterpret as problems due to interactions at the air-water interface.

Major remarks

1. The authors speculate about the better structural feature preservation in particles from 'clustering areas' (page 13) and suggest that these areas might be formed due to uneven glow-discharge on the surface.

The authors must provide a physical reasoning regarding what makes the plasma uneven at the described sub-micron scale. Another potential reason, however, is that the protein solution is partially dried out on the supporting film with only the clustered particles embedded in vitreous ice. For example, the micrographs in Figure 6 and Supplementary Figure 1 look like the graphene is only partially wetted and some regions are dried out. This non-uniform wetting may occur if the graphene surface is not sufficiently hydrophilic due to poor glow discharge or is severely contaminated with other species. The non-clustered protein particles may then be dried onto the surface without hydration and therefore have their structure disrupted, while in the cluster some amount of ice is preserved. The authors must show that this is not the case. From the tomographic cross-sections shown in Supplementary Figure 6 it is not obvious that the clusters localize to the graphene surface and the remaining particles to the air-water interface. If the authors cannot provide direct evidence, the speculation that the SA particles attached to the graphene are better structurally preserved (page 14) must be moved to the Discussion section or removed completely.

We understand the reviewer's concern and agree with the reviewer that it is important to understand the nature of protein's interaction on graphene surface. Our electron tomography analysis of the specimen excluded the possibility that the graphene was partially dried out. It is demonstrated clearly in the supplementary Movie 2 and Movie 3 that the ice thickness is reasonably even on the graphene surface and the particles distributed on both the graphene surface and the air-water interface. The fact that particles were uniformly dispersed on the air-water interface in the tomographic reconstruction is a strong evidence that the protein solution covers the entire graphene surface (Movie S2, S3). If the "lacuna area" on the graphene surface were dried out, we would not be able to detect particles on the air-water

interface just above it. The phenomenon of particle clustering on the graphene surface might be caused by the uneven hydrophilicity of the graphene surface because we used a rather mild glow-discharge treatment. Another possibility is that the filter paper peeled the particles away from the graphene surface and left a mark of the filter paper's fiber pattern on the grid. This of course is quite speculative and needs further investigation in future studies.

Our new analysis of the two separated subsets of particles based on their locations on the graphene grid have strengthened our argument that particles staying on the graphene-water interface are better preserved than those absorbed to the air-water interface as shown in the new Figure 6, S8 and S9 of the revised manuscript. We hope that the reviewer would agree with us on this statement with the new results in the revision.

2. The authors mention the use of a Cs corrector in this work. They should comment why they think spherical aberrations are a limiting factor in cryo-EM of small proteins and why a Cs corrector was necessary. Otherwise, the authors must make it clear that the Cs corrector is not a central part of this experiment, i.e. the same results are achievable without one. Failure to clarify this point may lead the field to misconceptions regarding the necessity of Cs-corrected TEMs for cryo-EM.

We have answered this to our response to Reviewer #1's first comment and #2's 6th comment. We have revised our manuscript to add a discussion of the Cs-corrector's role and explained that we do not intend to claim the necessity of a Cs-corrector for high-resolution structural determination.

3. The authors state that single-crystalline monolayer graphene was used in this study (page 5) but no data is provided to demonstrate that the graphene was indeed a single crystal and monolayer. The authors should experimentally characterize the graphene to support this statement (eg. as in Meyer et al, Nature 2017) or remove it from the text.

Thanks for the reviewer's comments. We have included more details about the production of the graphene-coated grids in the Materials and Methods section of our revised manuscript. We have also followed the reviewer's suggestion to examine the Fourier transform of micrographs taken on the same grid at different regions. They consistently showed very similar hexagonal lattice diffraction pattern in the same orientation, indicating the presence of a single-crystalline graphene film over the grid. We have updated our manuscript with this new result.

4. In the discussion section (page 15) the authors attempt to list the potential benefits of using graphene support films. However, there is no evidence to support statement (1) that the graphene reduces ice thickness, and (2) that the graphene reduces charging, radiation damage, and motion. The authors must provide supporting evidence for these claims or remove them. While the tomograms in Supplementary Figure 6 do indeed show that ice thickness of 200 – 500Å was achieved, this is also possible without the use of supporting films. To demonstrate reduced radiation damage and motion, the authors must provide B-factor plots and motion statistics. It is highly recommended to provide these in any case when reporting a cryo-EM structure.

We thank for the reviewer's advice.

1. For the graphene reducing ice thickness.

The tomogram #1 of 50 nm ice thickness demonstrates a relative thick area of the

specimen that we did not collect data. Regions like the one shown by tomogram #2 (10-20 nm) were selected for data collection. A general screening of sample quality in different types of grid and method was discussed in (Noble et al, 2018). According to the statistics from the Figure 3 in the Noble's paper, for cryo-samples prepared with holy carbon grid using general blotting method, a 56 nm averaged thickness at the hole center and a 99 nm averaged thickness at ~100 nm from the edge of the holes were reached. When using the Spotiton method with gold grids, they can get ice with 30 nm averaged thickness at the hole center and 61 nm averaged thickness at ~100 nm from the edge of the holes. Both methods introduced a sunken center with thinner ice than the edge of the hole. This work suggested the difficulty to make ice with uniform thickness below 30 nm on a holey grid without supporting film.

In our case of using the graphene-coated grids, we can get uniform ice thickness in the entire hole thinner than 30 nm quite robustly as revealed by the electron tomographic reconstruction of the specimen. But we do not exclude the other options in reducing the ice thickness without graphene supporting film, such as using very high concentration of proteins and adding detergent.

2. For the graphene reducing charging, radiation damage, and motion.

Graphene has a better conductivity than vitreous ice and amorphous carbon thus can compensate the accumulated charge faster. Previous studies by Russo and Passmore already characterized the graphene film's performance on cryo-EM specimen including the reduction of charging effect and radiation-induced particle motion. [Russo C J, Passmore L A. Controlling protein adsorption on graphene for cryo-EM using low-energy hydrogen plasmas[J]. *Nature Methods*, 2014, 11(6): 649.] We have cited the reference in the revised manuscript.

We removed the "graphene reduces radiation damage" from the manuscript as this does not have supporting evidence.

5. It is also highly recommended that the authors provide the orientation distributions of the particles on the graphene, as calculated via the 3D refinement. The authors may also wish to comment whether they attempted at all to solve these structures without the use of a supporting graphene film, as this will help elucidate the importance of the graphene support in this work.

We followed the reviewer's suggestion to provide the orientation distribution in Figure S3 and S4.

We tried to prepare cryo-EM specimens of streptavidin without graphene supporting film but failed to get good specimens for cryo-EM data collection. This emphasized the importance of using graphene-coated grids in the structural determination of streptavidin. We have put this comment in the revised manuscript.

6. The authors conclude the section 'Focused classification analysis of the biotin binding pocket of SA' with the statement 'The results above proved the capability of heterogeneity analysis for ligand binding detection of small proteins by single particle cryo-EM'. However, the specific case demonstrated here incorporated roughly equal number of particles with and without a ligand bound, which may not be the case in general. The authors may investigate this further (eg. whether a

minority ligand-bound class can be separated out) or alternatively edit/remove the above generalizing statement. The same applies to the related statement in the Discussion (page 15, beginning 'One may solve the structures...') since the authors do not demonstrate in this work that different ligand-occupied states can be classified. The authors should also cite other recent studies where ligand-bound and unbound states of proteins were separated by cryo-EM.

We followed the reviewer's suggestion to investigate further the classification's ability in separating the ligand-bound and unbound states. We generated multiple mixed datasets of asymmetric particles to mimic various ligand occupancy conditions and performed 3D reconstructions of them. We demonstrated that the classification could separate the ligand-binding state from the apo state to reveal the location of the ligand. We have updated our manuscript with the new analysis results.

We have cited recent studies using focused classification to separate ligand-bound and unbound states in the revision.

7. Further in the discussion section (page 15), the authors say that it was unexpected to find the particles in the specimen localize to the interfaces. However, this has been known to be the case since the very early days of cryo-EM (Dubochet, 1985) and more recent tomographic studies demonstrate the phenomenon clearly (Noble et al, 2018). This statement should be edited accordingly (eg. 'In agreement with other studies we found that the SA particles localize to the interfaces...') and the following references should be added: Noble et al, eLife (2018) and Dubochet et al, Trends in Biochem Sci (1985).

We thank for the reviewer's advice, and we have revised the manuscript as suggested.

8. The authors also claim that the problem of absorption to the air-water interface is more severe for smaller proteins (page 3-4) because the ice is thinner, which is incorrect, so this statement should be removed.

We have revised the manuscript as suggested.

Minor remarks

9. Any claims regarding the smallest protein structure solved using cryo-EM with the aid of certain methods (eg. phase plate, supporting films) should be carefully revised to match the current state of the field. The authors should consider citing Herzik et al (bioRxiv, 2018) who report high-resolution structures of 82 kDa and 64 kDa proteins by cryo-EM without the use of support films, a phase plate or aberration correctors. The authors should also cite Herzik et al (Nature Methods, 2017) for the structure determination of the 150 kDa enzyme aldolase using a TEM operated at 200 kV.

We have revised the manuscript as suggested.

10. The authors mention that high-pass filtering the particles (~50 Å in size) to 120 Å turned out to be necessary for their alignment and show that this improved the 2D classes (Supplementary Figure 2). The relevance of this filtering with frequency more than twice bigger than the particle size requires clarification. Why does it improve the alignment? For example, was this used to remove uneven background due to the supporting graphene film? This is likely to be the case if the graphene surface was not properly wetted as mentioned above, see eg. Fig. 23 in Dubochet et al, 1988.

Thanks for the reviewer's comments. In our experience, the 120 Å high-pass filtering clearly

improved our 2D classification results than non-filtering. This filtering removed the strong background bias at low frequency boosted by the phase plate.

11. The authors should clarify the duration of the glow-discharge treatment of the graphene-coated grids (page 17). In the same section, the authors may like to clarify why the blotting settings were different for the apo-SA and biotin-bound SA samples, given that the same grids and same concentration samples were used.

We have added the conditions for glow-discharge treatment of the grids in the Materials and Methods in the revision. We actually tried different blotting settings on both types of specimen. The settings that we put in the text were the ones that generated good grids for us to perform data collection.

12. The authors say that particles reach the interfaces 'quite fast' (page 16); they may refer to Glaeser (Curr Opin in Colloid Interface Sci, 2018) and the Supplement of Naydenova & Russo (Nature Comm, 2017), for a quantitative estimate of exactly how fast this occurs.

We have revised the manuscript as suggested.

13. Acronyms for the air-water interface (AWI) and the graphene-water interface (GWI) are not widely used and only make the text less readable, therefore it is recommended that these are removed.

Due to the word limits of the manuscript, we felt it is better to keep these abbreviations in the text.

14. The manuscript requires major corrections for spelling and grammar, which can significantly improve the readability of the text. For example, on page 17 'plugged' should read 'plunged', etc.

We have revised the language of the manuscript carefully.

Reviewers' Comments:

Reviewer #1:

Remarks to the Author:

The authors have improved the manuscript, and the work can, in principle, eventually be published. There are further issues that the authors will need to address beforehand.

The authors say: "The application of VPP is one of the most efficient ways". I disagree. VPP are not efficient and are notoriously difficult to use. Even the very developers of the technology have more recently abandoned their use, at least for the current generation VPPs.

Remove the schematic from Figure S1, it is confusing. Why is micrograph #0001 so astigmatic?

Orientation distributions in Figure S3 and S4 should be shown in identical configurations. Currently, the second preferred orientation is not aligned.

The amount of biotin occupancy needs to be quantified for figure 4. Also, focused classification should be performed with random starting points multiple times, and the results of biotin quantification shown be shown with error bars.

Colors are confusing in Figure 4C-D

When the authors say: "The 39 kDa trimeric dataset generated correct shapes and features in all different views" I assume they mean "both views", since there are only two displayed.

The manuscript should still be carefully proofread for grammar, punctuation, etc., eg:

"efficiency of extracting signals" should be "... signal"

"it has become more and more" – colloquial, consider revising

"to solve high-resolution structure" should be "to solve a high-resolution structure"

"under the conventional transmission..." should be "using conventional transmission..."

"mono-dispersed particles" should be "monodisperse particles"

... etc.

Reviewer #2:

Remarks to the Author:

The authors answered all questions and followed all suggestions.

Radostin Danev

Reviewer #3:

Remarks to the Author:

We are pleased that the authors have attempted to address many of the referees concerns but we still feel that the paper still has several problems and factual errors that are unresolved. In addition, while some of the revisions have improved the paper, others have actually made it worse and raised further questions.

If we focus just on the title and abstract, we find the following:

The title is correct:

The authors report near-atomic structures of streptavidin and biotin-bound streptavidin obtained by cryoelectron microscopy.

In the abstract:

Sentences 1 & 2 begin with true statements but include arbitrary numbers and misleading reasoning for their choice, i.e. the cut-off is determined by low contrast.

Sentences 3 & 4 are completely correct, they have determined these structures, but either the structures themselves, nor any of these individual methods used to determine them is novel. The authors claim that the combination of these techniques enables them to solve the smallest-to-date protein structure by cryo-EM and attempt to make some observations regarding the effect of the air-water interface on the specimen.

Sentence 5 is questionable at best and not supported by the paper.

Sentence 6 is obviously false – there are thousands of structures in the EMDB determined where all the particles were at the air water interface.

The major effort taken by the authors in response to the reviewers' comments was to extend the computational analysis of their data. However, in doing this they do not provide satisfactory responses to the rest of the major issues regarding their experiment, raised in the original reviews:

The authors continue to talk about uneven hydrophilicity of the graphene and uneven glow discharge. There is no physical reasoning provided for why this could be the case on sub-micron length scales. The authors mention the possibility of observing the footprint of the filter paper, this speculation is wrong by orders of magnitude. It seems more likely that the authors are actually seeing the effect of some contamination on the graphene surface which makes their observations unlikely to be reproducible.

The authors continue to claim that the particles at the air-water interface are less well preserved than those on the graphene surface. This claim is supported by Supplementary Fig. 8 showing a reconstruction from the two separate sets of particles. The authors report the observation of significant preferred orientation in the dataset composed of the particles on the air-water interface, so to make any further claims they must show that this is not the sole reason that degrades the resolution (together with the 3-fold smaller number of particles). The method of assignment of the particles to these subsets in the first place is error-prone. The authors still do not provide a satisfactory explanation to the question why the 'worse' particles look subjectively better on the micrographs.

Interestingly, the authors show the orientation-distributions of the apo-SA and biotin-SA particles (Supplementary Fig. 3 and 4) and these turn out to differ significantly. It is not clear why would the presence of the ligand in an internal binding site alter the orientation distribution of the protein particles on the surface. More likely, this difference is due to surface variations between the two different grids, posing the question of reproducibility again.

The authors still cannot claim that they demonstrate that the graphene reduces motion and/or charging unless they provide B-factor plots and motion statistics, and a comparison with controls, which they did not do in response to the review; they instead cited other work that is related.

In the section where the authors address the questions about the Cs corrector, they have stated that they do not claim its necessary (p18) but have at the same time added another claim that was not in the original manuscript: "This would allow a relatively efficient imaging strategy with a stable microscope optical condition because we can skip the search, hole centering, and autofocusing steps during the data acquisition..." which is completely unfounded and incorrect. This

is just one example of an additional problem introduced by the revisions.

The authors do not provide evidence for monolayer single-crystal graphene, as requested. The Fourier transforms showing the diffraction spots are misleadingly assigned to the four corners of a schematic grid; the authors must report the real crystal size instead. Are these just four neighbouring holes or holes in the same grid square or in different squares? Supplementary Figure 1 clearly shows that at least two different crystal orientations were observed. Furthermore, the presence of six diffraction spots in the FT does not exclude the option of having multilayer graphene composed of layers with the same lattice orientation. Therefore the authors cannot make any claims regarding the graphene being a monolayer.

In summary, the authors have correctly determined the structures in the paper and these are ok. But the reproducibility of the reported observations and their interpretation of the role of the various methods used remains a major concern. All of the claims regarding the benefits of using graphene/VPP/Cs correctors can only be adequately supported given appropriate control experiments or a detailed explanation of the reasoning for excluding alternate possibilities. They have not provided these.

Our response to the reviewers' comments are in **bold font**.

Reviewers' comments:

Reviewer #1 (Remarks to the Author):

The authors say: "The application of VPP is one of the most efficient ways". I disagree. VPP are not efficient and are notoriously difficult to use. Even the very developers of the technology have more recently abandoned their use, at least for the current generation VPPs.

We took the reviewer's suggestion and make sure there is no such claim in our revised manuscript.

Remove the schematic from Figure S1, it is confusing. Why is micrograph #0001 so astigmatic?

We removed the schematic in the revision. We noticed the high astigmatism in micrograph #0001. This is likely to be VPP-position-dependent astigmatism. But it does not affect the image processing using Gctf and Relion.

Orientation distributions in Figure S3 and S4 should be shown in identical configurations. Currently, the second preferred orientation is not aligned.

We have revised the figure as suggested.

The amount of biotin occupancy needs to be quantified for figure 4. Also, focused classification should be performed with random starting points multiple times, and the results of biotin quantification shown be shown with error bars.

We have followed the reviewer's suggestion to perform the classification with random starting points multiple times. The biotin occupancy statistics of apo-state and biotin-bound state were calculated and reported in the revised manuscript.

Colors are confusing in Figure 4C-D

We have revised Figure 4D with clearer color theme.

When the authors say: "The 39 kDa trimeric dataset generated correct shapes and features in all different views" I assume they mean "both views", since there are only two displayed.

We have revised the text accordingly.

The manuscript should still be carefully proofread for grammar, punctuation, etc., eg:

"efficiency of extracting signals" should be "... signal"

"it has become more and more" – colloquial, consider revising

"to solve high-resolution structure" should be "to solve a high-resolution structure"

"under the conventional transmission..." should be "using conventional transmission..."

"mono-dispersed particles" should be "monodisperse particles"

... etc.

We have used Nature research editing service to proofread the entire manuscript.

Reviewer #2 (Remarks to the Author):

The authors answered all questions and followed all suggestions.

Thanks to the reviewer to support the work.

Reviewer #3 (Remarks to the Author):

Sentences 1 & 2 begin with true statements but include arbitrary numbers and misleading reasoning for their choice, i.e. the cut-off is determined by low contrast.

We have removed the statement “mainly due to the low contrast of the molecules embedded in vitreous ice” from these sentences.

Sentences 3 & 4 are completely correct, they have determined these structures, but either the structures themselves, nor any of these individual methods used to determine them is novel. The authors claim that the combination of these techniques enables them to solve the smallest-to-date protein structure by cryo-EM and attempt to make some observations regarding the effect of the air-water interface on the specimen.

Sentence 5 is questionable at best and not supported by the paper.

This is the reason that we stated “has the potential”.

Sentence 6 is obviously false – there are thousands of structures in the EMDB determined where all the particles were at the air water interface.

We have revised the sentence as below:

Furthermore, we find that avoiding the adsorption of proteins onto the air-water interface is helpful to maintaining the high-resolution structural information.

The major effort taken by the authors in response to the reviewers' comments was to extend the computational analysis of their data. However, in doing this they do not provide satisfactory responses to the rest of the major issues regarding their experiment, raised in the original reviews: The authors continue to talk about uneven hydrophilicity of the graphene and uneven glow discharge. There is no physical reasoning provided for why this could be the case on sub-micron length scales. The authors mention the possibility of observing the footprint of the filter paper, this speculation is wrong by orders of magnitude. It seems more likely that the authors are actually seeing the effect of some contamination on the graphene surface which makes their observations unlikely to be reproducible.

We agree with the reviewer that we don't have a satisfactory explanation of the non-uniform distribution of particles on the graphene surface. We have revised our manuscript in the discussion section to acknowledge this and listed possible causes including the heterogeneous hydrophilicity, adsorbates contamination or more complicated graphene-water interfacial interactions on the graphene as reported in some literatures [(Ke, Peigen et al. 2010, Ko, Hsu et al. 2016)].

The authors continue to claim that the particles at the air-water interface are less well preserved than those on the graphene surface. This claim is supported by Supplementary Fig. 8 showing a reconstruction from the two separate sets of particles. The authors report the observation of

significant preferred orientation in the dataset composed of the particles on the air-water interface, so to make any further claims they must show that this is not the sole reason that degrades the resolution (together with the 3-fold smaller number of particles). The method of assignment of the particles to these subsets in the first place is error-prone. The authors still do not provide a satisfactory explanation to the question why the ‘worse’ particles look subjectively better on the micrographs.

To better address the effect of air-water interface on the structure of SA molecules, we prepared cryo-EM specimens of apo-SA on regular holey carbon grids without graphene support. Single particle analysis of these SA molecules demonstrated a strong preferential orientation, which is similar to the results from subset B. The particles cannot yield correct 3D structure at all.

To address the question of different number of particles in the datasets, we randomly selected the same number of particle from subset A (9547 from 27315), and performed the 3D refinement from scratch. We can still obtain a reconstruction with correct structural feature albeit at lower resolution (See Figure below). In contrast, the same number of particles from subset B cannot yield correct structure in global refinement (Figure below).

Randomly selected 9547 particles from subset A (4.3 Å)

9547 particles from subset B

These results indicate that the particles at the two interfaces do have different qualities. The preferred orientation in the top view of SA at the air-water interface has relatively more density along the projection direction than the other views. This may be one reason that they appeared better. Of course this needs to be studied more carefully in the future.

Interestingly, the authors show the orientation-distributions of the apo-SA and biotin-SA particles (Supplementary Fig. 3 and 4) and these turn out to differ significantly. It is not clear why would the presence of the ligand in an internal binding site alter the orientation distribution of the protein particles on the surface. More likely, this difference is due to surface variations between the two different grids, posing the question of reproducibility again.

We also noticed this orientation-distribution difference between apo-SA and biotin-SA datasets. Please be noted that we have 4 mM biotin molecule in the solution of biotin-SA specimen. We cannot rule out the effect of free biotin molecule on the graphene surface and air-water interface and therefore influence the orientation distribution of the particles. We have examined multiple specimens of apo-SA and biotin-SA made from different batches of

the graphene-coated grids. The quality of the specimens was reproducible.

The authors still cannot claim that they demonstrate that the graphene reduces motion and/or charging unless they provide B-factor plots and motion statistics, and a comparison with controls, which they did not do in response to the review; they instead cited other work that is related.

We now include B-factor plots for both apo-SA and biotin-SA datasets on the graphene in our revised manuscript. We also tried to perform single particle reconstruction of the apo-SA without graphene but failed due to the strong preferential orientation (Figure S10 in revised manuscript). Therefore we cannot use the dataset of SA without graphene as a control to calculate the B-factor plot. Instead, we collected single particle datasets of apo-ferritin with and without graphene to address this question. We used Relion particle polishing program to analyze full-frame aligned movie particle to estimate mainly the local motion of individual particles (See figure below).

In our results, the estimated B-factor of apo-ferritin on graphene is smaller than without graphene (A and B). The distribution of local motion speed (represented by the slope of particle motion) of apo-ferritin on graphene represent a smaller mean value and standard deviation than without graphene, indicating a reduced local motion. We are in the process to

analyze more datasets of different specimens with and without graphene and hope to report this in the future.

In the section where the authors address the questions about the Cs corrector, they have stated that they do not claim its necessary (p18) but have at the same time added another claim that was not in the original manuscript: “This would allow a relatively efficient imaging strategy with a stable microscope optical condition because we can skip the search, hole centering, and autofocusing steps during the data acquisition...” which is completely unfounded and incorrect. This is just one example of an additional problem introduced by the revisions.

We have removed the statement in the revision.

The authors do not provide evidence for monolayer single-crystal graphene, as requested. The Fourier transforms showing the diffraction spots are misleadingly assigned to the four corners of a schematic grid; the authors must report the real crystal size instead. Are these just four neighbouring holes or holes in the same grid square or in different squares? Supplementary Figure 1 clearly shows that at least two different crystal orientations were observed. Furthermore, the presence of six diffraction spots in the FT does not exclude the option of having multilayer graphene composed of layers with the same lattice orientation. Therefore the authors cannot make any claims regarding the graphene being a monolayer.

We are sorry for the confusing schematic figure. We wanted to clarify that the four micrographs were from different squares because we sequentially added acquisition position from each square (~ 500-600 position in each square). To avoid choosing micrographs from nearby holes, we selected micrographs from biotin-SA dataset (3309 micrographs) with enough number spacing (~ 1000). The reviewer was probably confused by the two different crystal orientations in Figure S1A-C and S1D. In fact, the Figure S1A-C were from micrographs of apo-SA grid and the Figure S1D were from micrographs of one biotin-SA grid. We are sorry that this was not clear in our previous version. In the revised manuscript, we have made it more clear in the legend of Figure S1.

We agree with the reviewer that the FT pattern cannot exclude the possibility of a single crystal with multiple layers. In our (coauthor Jin-Can Zhang and Hailin Peng) previous work to report the preparation of the single crystal graphene grids [Zhang, J. et al. Single Crystals: Clean Transfer of Large Graphene Single Crystals for High-Intactness Suspended Membranes and Liquid Cells, *Advanced Materials* 29 (2017)], we have characterized the crystal size of the graphene as sub-centimeter and the monolayer property of the graphene grids with Raman spectroscopy. The following figure of the Raman spectra was from Figure S7 of the *Advanced Materials* article. It shows that the no-polymer transfer method we used in the graphene grid preparation demonstrate a relatively high I_{2D}/I_G ratio (6–8) along with a small full width at half-maximum (FWHM) of 2D band (24 cm^{-1}) indicating a clean monolayer graphene crystal.

In summary, the authors have correctly determined the structures in the paper and these are ok. But the reproducibility of the reported observations and their interpretation of the role of the various methods used remains a major concern. All of the claims regarding the benefits of using graphene/VPP/Cs correctors can only be adequately supported given appropriate control experiments or a detailed explanation of the reasoning for excluding alternate possibilities. They have not provided these.

We have revised the manuscript with more conserved statements specific to our case study.

Reviewers' Comments:

Reviewer #1:

Remarks to the Author:

What is the occupancy of biotin for individual classes? The authors should put this information in the figure, quantifying the occupancy with error bars.

The authors need to make it explicitly clear that the current focused classification approach for quantifying ligand density is limited, and that full occupancy of the bound inhibitor cannot be obtained. Some commentary on the reason for this is warranted.

Reviewer #3:

Remarks to the Author:

The authors have addressed some of our queries in the initial review, but problems remain:

(1) On page 17, line 350, the authors claim 'The above results collectively indicate a better preserved native structure of the SA on the GWI than those molecules adsorbed on the AWI'. The reviewer considers this as an erroneous interpretation of the experiments. The experimental evidence suggests that the major difference between the two particle sets is the orientation distribution, not the structural integrity of the particles. The same applies to the unsupported control specimen. Orientation distribution with insufficient coverage of Fourier space is known to prevent high-resolution structure determination. This statement should be removed. Accordingly, the last sentence of the abstract must be removed, as the claim is unsupported.

(2) In response to the reviewer's concern regarding claims that the graphene reduces motion and/or charging, the authors show a comparison for apoferritin (which is not included in this paper) and say they are in the process of analyzing more datasets. The reviewer strongly encourages this further work; however, the speculative statements regarding the graphene reducing the movement of the SA particles are not supported by this paper and should be removed (page 19, lines 395-396).

(3) In response to our remark about the different orientation distributions of the apo-SA and biotin-bound SA, the authors point out the possibility of the effect of free biotin on the surface interactions. This should be described in the discussion section of the paper.

(4) In response to the reviewer's concerns regarding reproducibility, the authors say they examined multiple specimens from different batches. The authors should specify the number of specimens and batches in the paper. The authors state that 'the quality of the specimens was reproducible'. What about the orientation distribution of the particles?

*****If the authors remove the erroneous statements listed in (1) and (2), and address our comments (3) and (4) by appropriate changes to the main text, the paper can be published as a case report of a small-protein structure determination by cryo-EM, which the authors have successfully done. The unsubstantiated claims regarding improvement due to various methods employed must be removed.*****

Our response to the reviewers' comments are in **bold font**.

Reviewers' comments:

Reviewer #1 (Remarks to the Author):

What is the occupancy of biotin for individual classes? The authors should put this information in the figure, quantifying the occupancy with error bars.

The occupancy of biotin for individual classes with error bars are now clearly labelled in Figure 4.

The authors need to make it explicitly clear that the current focused classification approach for quantifying ligand density is limited, and that full occupancy of the bound inhibitor cannot be obtained. Some commentary on the reason for this is warranted.

We have added a commentary in the Discussion to explicitly state the limitation of current focused classification approach and the possible reason as below (page 18).

“Our focused classification approach to quantify the ligand density of the dataset suggested a limited capability to obtain a full occupancy of the bound ligand. This is likely due to the low SNR of the images and the current algorithm’s limitation.”

Reviewer #3 (Remarks to the Author):

The authors have addressed some of our queries in the initial review, but problems remain:

(1) On page 17, line 350, the authors claim "The above results collectively indicate a better preserved native structure of the SA on the GWI than those molecules adsorbed on the AWI. The reviewer considers this as an erroneous interpretation of the experiments. The experimental evidence suggests that the major difference between the two particle sets is the orientation distribution, not the structural integrity of the particles. The same applies to the unsupported control specimen. Orientation distribution with insufficient coverage of Fourier space is known to prevent high-resolution structure determination. This statement should be removed. Accordingly, the last sentence of the abstract must be removed, as the claim is unsupported.

We abide the reviewer’s point. We have removed the two sentences from the Results section and the abstract as requested by the reviewer. We also made several minor revisions in the Discussion section about the protein molecules on the AWI to reflect the reviewer’s point.

(2) In response to the reviewer's concern regarding claims that the graphene reduces motion and/or charging, the authors show a comparison for apoferritin (which is not included in this paper) and say they are in the process of analyzing more datasets. The reviewer strongly encourages this further work; however, the speculative statements regarding the graphene reducing the movement of the SA particles are not supported by this paper and should be removed (page 19, lines 395-396).

We have removed this sentence from the text.

(3) In response to our remark about the different orientation distributions of the apo-SA and biotin-bound SA, the authors point out the possibility of the effect of free biotin on the surface interactions. This should be described in the discussion section of the paper.

We have added the related comments in the Discussion part as below (page 19).

“We also noticed that the orientation distributions of apo-SA and biotin-SA were slightly different. This difference is probably an effect on the surface property of GWI by 4 mM free biotin molecule in the biotin-SA solution.”

(4) In response to the reviewer's concerns regarding reproducibility, the authors say they examined multiple specimens from different batches. The authors should specify the number of specimens and batches in the paper. The authors state that 'the quality of the specimens was reproducible'. What about the orientation distribution of the particles?

We have used two apo-SA grids and four biotin-SA grids prepared in two batches 40 days apart from each other. The quality of the specimens was reproducible for both particle distribution in the specimen and orientation distribution of the particles. We have added a statement in the Discussion part to meet the request from the reviewer as below (page 19).

“In the total two apo-SA grids and four biotin-SA grids prepared in two batches 40 days apart from each other, we found that the distribution of particles in the specimens and orientations as well as the image quality were reproducible in our hands.”

*****If the authors remove the erroneous statements listed in (1) and (2), and address our comments (3) and (4) by appropriate changes to the main text, the paper can be published as a case report of a small-protein structure determination by cryo-EM, which the authors have successfully done. The unsubstantiated claims regarding improvement due to various methods employed must be removed.*****

We have revised the manuscript explicitly as suggested by the reviewer.